# Optical control of competing exchange interactions and coherent spin-charge coupling in two-orbital Mott insulators

**Marion M. S. Barbeau[1]⋆, Martin Eckstein[2],**
**Mikhail I. Katsnelson[1] and Johan H. Mentink[1]**

**1** Institute for Molecules and Materials, Radboud University Nijmegen,
Heyendaalseweg 135, NL-6525 AJ Nijmegen, The Netherlands.
**2** Friedrich-Alexander Universität Erlangen-Nürnberg, Institut für Theoretische Physik,
Standtstraße 7, 91058 Erlangen, Germany.

⋆ m.barbeau@science.ru.nl

## Abstract

In order to have a better understanding of ultrafast electrical control of exchange interactions in multi-orbital systems, we study a two-orbital Hubbard model at half filling under the action of a time-periodic electric field. Using suitable projection operators and a generalized time-dependent canonical transformation, we derive an effective Hamiltonian which describes two different regimes. First, for a wide range of non-resonant frequencies, we find a change of the bilinear Heisenberg exchange $J_{ex}$ that is analogous to the single-orbital case. Moreover we demonstrate that also the additional biquadratic exchange interaction $B_{ex}$ can be enhanced, reduced and even change sign depending on the electric field. Second, for special driving frequencies, we demonstrate a novel spin-charge coupling phenomenon enabling coherent transfer between spin and charge degrees of freedom of doubly ionized states. These results are confirmed by an exact time-evolution of the full two-orbital Mott-Hubbard Hamiltonian.



# 1  Introduction

The exchange interaction $J_{\mathrm{ex}}$ between microscopic spins is the strongest interaction in magnetic systems. Therefore, the control of exchange is a very promising way for ultrafast control of magnetic order, with potentially high energy efficiency. Recently, the ultrafast control of $J_{\mathrm{ex}}$ has received significant interest both in experiments with cold atoms as well as in condensed matter systems [1–11]. An appealing way to achieve a control of $J_{\mathrm{ex}}$ is to use periodic driving with off-resonant pulses as was extensively investigated theoretically [12–17]. In particular, it was predicted theoretically [12,13] and recently confirmed experimentally [11] that by tuning the strength and frequency of the driving, $J_{ex}$ can be reduced, enhanced and even reverse sign in a reversible way. However, so far, most theoretical studies rely on single-orbital models, while multi-orbital physics is important in many materials. Moreover, existing studies [18–21] on multi-orbital systems did not reveal the role of orbital dynamics on the control of exchange interactions.

In order to have a better understanding of the influence of orbital dynamics on the ultrafast and reversible control of exchange, we report the study of a two-orbital system at half filling under the effect of a periodic electric field. There are two main differences between single and multi-orbital systems which are already captured in the two-orbital case. First, there is the Hund interaction $J_{\mathrm{H}}$ that directly arises from inter-orbital exchange on the same site. At half filling and for $J_{\mathrm{H}}{>}0$, each orbital is singly occupied and the low-energy degrees of freedom are spin-one states which interact both via a normal Heisenberg exchange $J_{\mathrm{ex}}\vec{S}_i{\cdot}\vec{S}_j$ and with a biquadratic exchange interaction $B_{\mathrm{ex}}(\vec{S}_i{\cdot}\vec{S}_j)^2$. While $J_{\mathrm{ex}}$ favors collinear spin order at neighboring sites, for $B_{\mathrm{ex}}{>}0$ , non-collinear spin order can become preferential. For calssical spins, the presence of biquadratic exchange interaction can lead to spin spiral states [22]. For quantum spins in low dimentions systems, the presence of $B_{\mathrm{ex}}$ can give rise to disordered phases such as dimerized or quadruolar phase [23–25]. Second, as illustrated in Figure 1, the two-orbital model has excited states which are doubly ionized and strongly gapped with respect to states with only one electron in each orbital (we will refer to configurations with one electron in each orbital as singly occupied states). The doubly ionized states are charge states, which are coupled to singly occupied states by two subsequent hopping processes.

Below we demonstrate that there exist two distinct regimes for the non-resonantly driven two-orbital model. First, a regime for which the control of intersite exchange interactions dominates. We recover a Heisenberg exchange interaction $J_{\mathrm{ex}}(\mathcal{E}, \omega)$ that is similar as in single-

orbital systems, where $\mathcal{E}$ is the driving strength and $\omega$ the driving frequency. We find that analogous to $J_{\text{ex}}(\mathcal{E}, \omega)$, also the biquadratic exchange interaction $B_{\text{ex}}(\mathcal{E}, \omega)$ can be reduced, enhanced and reverse sign by tuning the strength and frequency of the driving field. In addition, we find a regime for which the exchange interactions compete *i.e.* $J_{\text{ex}}(\mathcal{E}, \omega) \sim B_{\text{ex}}(\mathcal{E}, \omega) > 0$. Second, we elucidate a regime for which a new type of spin-charge coupling phenomenon dominates over the exchange interaction. In this regime, a reversible transfer between spin and charge degrees of freedom is feasible.

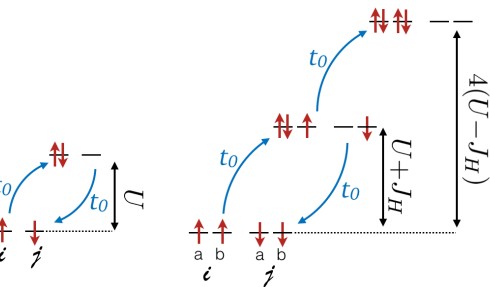

Figure 1: (Color online) Sketch of virtual hopping processes $t_0$ (in blue) between site $i$ and $j$ with different number of doublons $d$ in the case of a single orbital (left) and a two-orbital ($a$ and $b$) model (right). Small red arrows indicate the spins of electrons. $U$ denotes the Coulomb repulsion and $J_H$ is the on-site Hund exchange interaction.

The paper is organized as follows: in Section 2 we introduce the two-orbital Hubbard model, define projection operators, and introduce a generalization of the time-dependent canonical transformation [16, 18, 26, 27]. In Section 3 we derive the effective Hamiltonian, study its low energy part, and show how to map it onto a spin-one model. From this spin-one model, the Heisenberg exchange interaction as well as the additional biquadratic exchange interaction are extracted. Beyond the spin model, we study the spin-charge coupling phenomenon. Moreover, we confirm the analytical results on the spin-charge coupling by computing the time evolution of the full two-orbital Mott-Hubbard model for a two-site cluster. Finally, in Section 5, we draw conclusions.

## 2 Method

### 2.1 Electronic model

To study the role of orbital dynamics on the electrical control of exchange, we investigate a two-orbital model at half-filling. This can be associated with the $e_g$ band of an oxide compound. The Hamiltonian is given by $\hat{H}(t) = \hat{H}_U + \hat{H}_{\text{kin}}(t)$, where $\hat{H}_U = \hat{H}_{\text{nn}} + \hat{H}_{\text{sf}}$. $\hat{H}_{\text{nn}}$, $\hat{H}_{\text{sf}}$ and $\hat{H}_{\text{kin}}$ contain the density-density interaction, the spin-flip and pair hopping, and the intersite hopping, respectively:

$$\hat{H}_{\text{nn}} = \sum_i \sum_{\alpha \neq \beta, \sigma} \left\{ U \hat{n}_{i\alpha\uparrow} \hat{n}_{i\alpha\downarrow} + \frac{(U - 2J_H)}{2} \hat{n}_{i\alpha\sigma} \hat{n}_{i\beta\bar{\sigma}} + \frac{(U - 3J_H)}{2} \hat{n}_{i\alpha\sigma} \hat{n}_{i\beta\sigma} \right\} \tag{1}$$

$$\hat{H}_{\text{sf}} = -J_H \sum_{i, \alpha \neq \beta} \left( \hat{c}_{i\alpha\uparrow}^\dagger \hat{c}_{i\alpha\downarrow} \hat{c}_{i\beta\downarrow}^\dagger \hat{c}_{i\beta\uparrow} + \hat{c}_{i\alpha\uparrow}^\dagger \hat{c}_{i\alpha\downarrow} \hat{c}_{i\beta\downarrow}^\dagger \hat{c}_{i\beta\uparrow} \right) \tag{2}$$

$$\hat{H}_{\text{kin}}(t) = -\sum_{<i,j>} t_{ij}(t) \sum_{\alpha,\sigma} \hat{c}^{\dagger}_{i\alpha\sigma} \hat{c}_{j\alpha\sigma}. \tag{3}$$

Here $\hat{c}^{\dagger}_{i\alpha\sigma}(\hat{c}_{i\alpha\sigma})$ are fermionic creation (anihilation) operators for site $i$, orbital $\alpha = a, b$, spin $\sigma = \uparrow, \downarrow$, and $\hat{n}_{i\alpha\sigma} = \hat{c}^{\dagger}_{i\alpha\sigma} \hat{c}_{i\alpha\sigma}$. $U$ is the on-site Coulomb interaction and $J_{\text{H}}$ is the Hund exchange interaction.

The time-dependence of the hopping term originates from the external electric field which is described using the Peierls substitution $t_{ij}(t) = t_0 e^{ieA_{ij}(t)}$ [12, 28, 29], where $e$ is the electronic charge, $A_{ij}(t) = -\frac{1}{\omega} E_0 \cos(\omega t)(R_i - R_j)$ is the projection of the vector potential along the direction from site $i$ to $j$, where $E_0$ is the amplitude of the field. Since both $e_g$ orbitals originate from $d$ orbitals, no on-site electric dipole transition are allowed. We define the parameter $\mathcal{E} = eaE_0/\omega$ which represents the driving strength, whith $a = |R_i - R_j|$ and we take $t_0 = 1$ for the numerical calculations.

## 2.2 Projection operators

The conventional way to derive the exchange interaction is to use a canonical transformation also known as Schrieffer-Wolff transformation [16, 18, 26, 27]. For the two-orbital case, this is more involved due to the Hund interaction $J_{\text{H}}$. To deal with this additional complexity, we restrict the Hilbert space to blocks involving only two sites $(ij)$. For all states $|\phi_k\rangle$ on the bond $(ij)$, we then define projection operators $\hat{P}^{\gamma}_d(N, M)$ onto the following quantum numbers:

- Particle number:

$$(N - \hat{N}) \, \hat{P}^{\gamma}_d(N, M) \, |\phi_k\rangle = 0, \tag{4}$$

where $\hat{N} = \sum_{i\alpha\sigma} \hat{n}_{i\alpha\sigma}$ and $N = 0, ..., 8$ the number of electrons which occupy the system.

- Total spin $\hat{S}^z$ component:

$$(M - \hat{S}^z_{\text{tot}}) \, \hat{P}^{\gamma}_d(N, M) \, |\phi_k\rangle = 0, \tag{5}$$

where $\hat{S}^z_{\text{tot}} = \sum_i (\hat{S}^z_{ia} + \hat{S}^z_{ib})$ and $M = -2, ..., 2$.

Below we focus on a half filled system, $N = 4$. In addition, we consider an antiferromagnetic state such that $M = 0$, and write $\hat{P}^{\gamma}_d(N = 4, M = 0) \equiv \hat{P}^{\gamma}_d$.

- Number of doublons:

$$(d - \hat{d}) \, \hat{P}^{\gamma}_d \, |\phi_k\rangle = 0, \tag{6}$$

where $d = 0, 1, 2$ is the double occupancy and $\hat{d} = \sum_{i\alpha} \hat{n}_{i\alpha\uparrow} \hat{n}_{i\alpha\downarrow}$. Hence, $\hat{P}^{\gamma}_d$ projects onto states with $d$ doublons.

- Hund rule violation:

$$(\nu - \hat{\nu}) \, \hat{P}^{\gamma}_d \, |\phi_k\rangle = 0, \tag{7}$$

with $\hat{\nu} = \sum_{i\alpha \neq \beta} \frac{1}{2}(\hat{n}_{i\alpha\uparrow} \hat{h}_{i\alpha\downarrow} \hat{h}_{i\beta\uparrow} \hat{n}_{i\beta\downarrow} + \hat{n}_{i\alpha\uparrow} \hat{n}_{i\alpha\downarrow} \hat{h}_{i\beta\uparrow} \hat{h}_{i\beta\downarrow})$, where $\hat{h}_{i\alpha\sigma} = (1 - \hat{n}_{i\alpha\sigma})$. The value $\nu = 0, 1$ corresponds to configurations that satisfy or violate local spin alignment dictated by Hund exchange, respectively. For example, in the $\hat{P}^{\gamma}_0$ sector, the states with $\nu = 0$ are $|\uparrow, \uparrow\rangle_i |\downarrow, \downarrow\rangle_j$,

$|\downarrow,\downarrow\rangle_i|\uparrow,\uparrow\rangle_j$, and the $\nu{=}1$ states are $|\uparrow,\downarrow\rangle_i|\uparrow,\downarrow\rangle_j$, $|\downarrow,\uparrow\rangle_i|\downarrow,\uparrow\rangle_j$, $|\uparrow,\downarrow\rangle_i|\uparrow,\downarrow\rangle_j$, $|\downarrow,\uparrow\rangle_i|\uparrow,\downarrow\rangle_j$, where $|\sigma_a,\sigma'_b\rangle_i = \hat{c}^\dagger_{ib\sigma'}\hat{c}^\dagger_{ia\sigma}|0\rangle$.

Although $\left[\hat{H}_{\mathrm{sf}},\hat{P}^\nu_d\right]{=}0$, the states $\hat{P}^\nu_d|\phi_k\rangle$, with $\nu{=}1$ do not diagonalize $\hat{H}_{\mathrm{sf}}$. In principle, it is possible to further decompose $\hat{P}^\nu_d$ by introducing additional quantum numbers that project on states that simultaneously diagonalize $\hat{P}^\nu_d$ and $\hat{H}_{\mathrm{sf}}$. Here we restrict ourselves to the projectors $\hat{P}^\nu_d$, since this is already sufficient to describe the control of the biquadratic exchange interaction as well as the spin-charge coupling, as we discuss in more detail below.

It is shown in Appendix A that explicit expressions for $\hat{P}^\nu_d(N,M)$ in terms of single-electron operators can be derived using

$$\hat{p}(i) = \prod_{\alpha,\sigma}(\hat{n}_{i\alpha\sigma} + \hat{h}_{i\alpha\sigma}), \tag{8}$$

where $\hat{h}_{i\alpha\sigma}{=}(1{-}\hat{n}_{i\alpha\sigma})$. With these definitions, the identity reads

$$1 = \hat{p}(i)\hat{p}(j) = \sum_{d,\nu,N,M}\hat{P}^\nu_d(N,M). \tag{9}$$

The hopping term Eq. (3) connects $\hat{P}^\nu_d$ with different $d$ and can be re-written in terms of operators $\hat{T}^{+1}(t)$, $\hat{T}^{-1}(t)$ and $\hat{T}^0(t)$ that change $d$ by $+1$, $-1$ and $0$ respectively

$$\hat{H}_{\mathrm{kin}}(t) = \hat{T}^{+1}(t) + \hat{T}^{-1}(t) + \hat{T}^0(t), \tag{10}$$

where

$$\hat{T}^{+1}(t) = \sum_{\nu=0,1}\left(\hat{P}^\nu_2\hat{H}_{\mathrm{kin}}(t)\hat{P}^0_1 + \hat{P}^0_1\hat{H}_{\mathrm{kin}}(t)\hat{P}^\nu_0\right), \tag{11}$$

$$\hat{T}^{-1}(t) = \sum_{\nu=0,1}\left(\hat{P}^\nu_0\hat{H}_{\mathrm{kin}}(t)\hat{P}^0_1 + \hat{P}^0_1\hat{H}_{\mathrm{kin}}(t)\hat{P}^\nu_2\right), \tag{12}$$

and

$$\hat{T}^0(t) = \hat{P}^0_1\hat{H}_{\mathrm{kin}}(t)\hat{P}^1_1 + \hat{P}^1_1\hat{H}_{\mathrm{kin}}(t)\hat{P}^0_1. \tag{13}$$

Expressions for for $\hat{T}^{+1}(t)$, $\hat{T}^{-1}(t)$ and $\hat{T}^0(t)$ in terms of single electron operators are given in Appendix A. The projection operators $\hat{P}^\nu_d$ and hopping operators $\hat{T}^{+1}(t)$, $\hat{T}^{-1}(t)$, $\hat{T}^0(t)$ play an important role in the canonical transformation described below.

## 2.3  Generalized time-dependent canonical transformation

The canonical transformation is a technique which enables the derivation of an effective Hamiltonian for the subspace of states $\hat{P}^\nu_d$ [16, 18, 26, 27, 30]. Formally, this is achieved by unitary transformation $\hat{V}(t){=}e^{-i\hat{S}(t)}$ that transforms the Hamiltonian $\hat{H}(t)$ to a rotated frame. The effective Hamiltonian in the rotated frame reads

$$\hat{H}_{\mathrm{eff}}(t) = \hat{V}^\dagger(t)(\hat{H}(t) - i\partial_t)\hat{V}(t). \tag{14}$$

The aim is to identify a suitable subspace (defined by values of $d$ and $\nu$) and determine $\hat{V}$ such that $\hat{H}_{\mathrm{eff}}$ leaves this subspace invariant. To do this, we perform the unitary transformation

perturbatively, treating the hopping parameter $t_0 \ll U$ as a perturbation. We expand $i\hat{S}(t)$ and $\hat{H}_{\text{eff}}(t)$ in terms of a Taylor series

$$i\hat{S}(t) = \sum_{n=1}^{+\infty} i\hat{S}^{(n)}(t), \tag{15}$$

$$\hat{H}_{\text{eff}}(t) = \sum_{n=0}^{+\infty} \hat{H}_{\text{eff}}^{(n)}(t), \tag{16}$$

where $\hat{S}^{(n)}, \hat{H}_{\text{eff}}^{(n)} \propto t_0^n$. For deriving a pure spin model, one could construct the unitary transformation such that $\hat{H}_{\text{eff}}^{(n)}$ does not contain terms that change $d$ [16, 30–32], and obtain an effective Hamiltonian in the subspace $d = 0$. Here we use a more general requirement which will allow us to derive an effective Hamiltonian in a subspace different from that without doublons. This turns out to be crucial for a description of multi-orbital systems. We enlarge our effective model and keep terms that change $d$ by $\pm 2$, while we design $\hat{P}_d^{\gamma} i\hat{S}^{(n)} \hat{P}_{d'}^{\gamma'}$ such that

$$\hat{P}_d^{\gamma} \hat{H}_{\text{eff}}^{(n)}(t) \hat{P}_{d\pm 1}^{\gamma'} = 0. \tag{17}$$

At half filling and without inter-orbital hopping ($t_{\alpha \neq \beta}=0$), only odd orders of $i\hat{S}^{(n)}(t) \propto t_0^n$ remain,

$$i\hat{S}^{(n)}(t) = i\hat{S}^{(1)}(t) + i\hat{S}^{(3)}(t) + \mathcal{O}(t_0^5). \tag{18}$$

Eqs. (17) and (18) not only allow us to obtain an effective description of the low energy states $\hat{P}_0^{\gamma}$, but also enable us to keep track of the coupling between the low energy space (spin: $\hat{P}_0^{\gamma}$) and the space with the highest excited states (charge: $\hat{P}_2^{\gamma}$). Eqs. (17) and (18) yields the zeroth order contribution to the effective Hamiltonian

$$\hat{H}_{\text{eff}}^{(0)} = \hat{H}_{\text{nn}} + \hat{H}_{\text{sf}}. \tag{19}$$

Using the projection operators, we obtain the following equation for $i\hat{S}^{(1)}(t)$

$$\hat{P}_d^{\gamma} \big[ \hat{T}^{\pm 1}(t) + [i\hat{S}^{(1)}(t), \hat{H}_U] - \partial_t i\hat{S}^{(1)}(t) \big] \hat{P}_{d\pm 1}^{\gamma'} = 0. \tag{20}$$

In contrast to the zeroth order contribution $\hat{H}_{\text{eff}}^{(0)}$, Eq. (20) is a time-dependent equation. In principle, it is possible to solve this equation for arbitrary time-dependency, as worked out in [18]. Here we use a simpler algebraic solution that is feasible for time periodic driving and which is closely related to Floquet theory [12, 16, 31] and the high frequency expansion [13, 26, 32]. Given a time periodic electric field $E(t)=E(t + T)$ with a period $T=\frac{2\pi}{\omega}$, we can expand $\hat{T}^{\pm 1}(t)$ and $i\hat{S}^{(n)}(t)$ in a Fourier series as follows

$$\hat{T}^{\pm 1}(t) = \sum_{m=-\infty}^{+\infty} \hat{T}_m^{\pm 1} e^{im\omega t}, \quad i\hat{S}^{(n)}(t) = \sum_{m=-\infty}^{+\infty} i\hat{S}_m^{(n)} e^{im\omega t}, \tag{21}$$

where $m$ is the Fourier index, which can be seen as the number of virtual photons absorbed by the system [31]. Using Eqs. (20) and (21), we obtain:

$$\hat{P}_d^{\gamma} i\hat{S}_m^{(1)} \hat{P}_{d\pm 1}^{\gamma'} = C_{dd\pm 1}^{\gamma\gamma', m} \hat{P}_d^{\gamma} \hat{T}_m^{\pm 1} \hat{P}_{d\pm 1}^{\gamma'}, \tag{22}$$

with $C_{dd'}^{\gamma\gamma', m}=(E_d^{\gamma} - E_{d'}^{\gamma'} + m\omega)^{-1}$ and

$$\hat{P}_d^{\gamma} \hat{H}_U \hat{P}_{d'}^{\gamma'} = \delta_{dd'} \delta_{\gamma\gamma'} E_d^{\gamma} \hat{P}_d^{\gamma}. \tag{23}$$

For $\nu = 1$, $E_d^\nu$ is a matrix and we would have to further decompose $P_d^\nu$ for the procedure to be exact (see also Section 2.2). Here instead we use an approximation $E_d^\nu = \min(E_d^{\nu,\mu})$, where $E_d^{\nu,\mu}$ are the eigenvalues obtained from diagonalizing $\langle \phi_k | \hat{P}_d^\nu \hat{H}_U \hat{P}_d^\nu | \phi_{k'} \rangle$. This is a generalization of the energy approximation employed in [33], where $E_d$ is approximated by the mean energy of all states for given $d$. The present approximation is accurate for

$$|E_d^{\nu,\mu} - E_d^{\nu,\mu'}| \ll |E_d^\nu - E_{d'}^{\nu'}|, \tag{24}$$

where the number of doublons $d \neq d'$ and the Hund rule violation index $\nu \neq \nu'$. We find that this condition is satisfied for the calculations presented in Section 3.

The first order effective Hamiltonian $\hat{H}_{\text{eff}}^{(1)}(t)$ vanishes because $\hat{T}^0(t) = 0$ for orbital-diagonal hopping $t_{\alpha \neq \beta} = 0$. Eq. (22) allows us to compute higher order contributions to $\hat{H}_{\text{eff}}(t)$ in a straightforward way. The second order contribution reads

$$\hat{P}_d^\nu \hat{H}_{\text{eff}}^{(2)}(t) \hat{P}_{d'}^{\nu'} = \sum_m \sum_{k+l=m} \hat{P}_d^\nu \frac{1}{2} [i\hat{S}_k^{(1)}, \hat{T}_l^{\pm 1}] \hat{P}_{d'}^{\nu'} e^{im\omega t}, \tag{25}$$

where, $d, d' = 0, 2$ and $\nu, \nu' = 0, 1$.

The third order contribution to $\hat{H}_{\text{eff}}(t)$ gives us an expression for $\hat{P}_d^\nu i\hat{S}_m^{(3)} \hat{P}_{d\pm 1}^{\nu'}$:

$$\hat{P}_d^\nu i\hat{S}_m^{(3)} \hat{P}_{d\pm 1}^{\nu'} = C_{dd\pm 1}^{\nu\nu',m} \frac{1}{3} \sum_{p+q+r=m} \hat{P}_d^\nu [i\hat{S}_p^{(1)}, [i\hat{S}_q^{(1)}, \hat{T}_r^{\pm 1}]] \hat{P}_{d\pm 1}^{\nu'}. \tag{26}$$

This yields the following fourth order contribution to the effective Hamiltonian:

$$\hat{P}_d^\nu \hat{H}_{\text{eff}}^{(4)}(t) \hat{P}_{d'}^{\nu'} = \frac{1}{8} \sum_p \sum_{k+l+m+n=p} \hat{P}_d^\nu [i\hat{S}_k^{(1)}, [i\hat{S}_l^{(1)}, [i\hat{S}_m^{(1)}, \hat{T}_n^{\pm 1}]]] \hat{P}_{d'}^{\nu'} e^{ip\omega t}, \tag{27}$$

with Eqs. (25) and (27) we have derived the central result of this section, namely an effective Hamiltonian up to fourth order in the hopping.

# 3 Results

In this section we present the results obtained with the projection operators and the effective Hamiltonian derived above. First we show that the $d = 0$ part can be mapped onto an effective spin-one ($S = 1$) model. This requires two additional unitary transformations: one to reduce $\hat{H}_{\text{eff}}$ to the $d = 0$ sector and a second to specialize to the $S = 1$ states only. In the effective spin-one Hamiltonian, we extract the Heisenberg $J_{\text{ex}}(\mathcal{E}, \omega)$ and biquadratic $B_{\text{ex}}(\mathcal{E}, \omega)$ exchange interactions. Second, we focus on the coupling terms between sectors $d = 0$ and $d = 2$ by taking both of them into account in the low energy description. This goes beyond the spin model and captures the spin-charge coupling dynamics.

## 3.1 Spin-one model

According to condition Eq. (17), the full effective Hamiltonian yields

$$\sum_{n=2,4} \hat{H}_{\text{eff}}^{(n)}(t) = \sum_{\substack{n=2,4 \\ \nu\nu'}} \left\{ \hat{P}_0^\nu \hat{H}_{\text{eff}}^{(n)}(t) \hat{P}_0^{\nu'} + \hat{P}_2^\nu \hat{H}_{\text{eff}}^{(n)}(t) \hat{P}_2^{\nu'} + \hat{P}_0^\nu \hat{H}_{\text{eff}}^{(n)}(t) \hat{P}_2^{\nu'} + \hat{P}_2^\nu \hat{H}_{\text{eff}}^{(n)}(t) \hat{P}_0^{\nu'} \right\}. \tag{28}$$

In this subsection we study the low energy effective Hamiltonian up to fourth order in the hopping. In the derivation of the spin-one model, we have to consider the sector $\hat{P}_2^{\gamma}$ as a high energy sector and perform a second time-dependent canonical transformation in order to project out states for which $d=2$:

$$\hat{H}_{\text{eff}}^{d=0}(t) = \sum_{\gamma\gamma'} \hat{P}_0^{\gamma} \left\{ \hat{H}_{\text{eff}}^{(2)}(t) + \hat{H}_{\text{eff}}^{(4)}(t) + \tilde{H}_{\text{eff}}^{(4)}(t) \right\} \hat{P}_0^{\gamma'}, \tag{29}$$

where

$$\sum_{\gamma,\gamma'} \hat{P}_0^{\gamma} \tilde{H}_{\text{eff}}^{(4)}(t) \hat{P}_0^{\gamma'} = \sum_{\substack{\gamma'' \\ m,m'}} C_{20}^{\gamma'0,m} \hat{P}_0^{\gamma} \hat{H}_{\text{eff},m}^{(2)}(t) \hat{P}_2^{\gamma''} \hat{H}_{\text{eff},m'}^{(2)}(t) \hat{P}_0^{\gamma'}. \tag{30}$$

We used that $C_{d0}^{\gamma0,m} = C_{d0}^{\gamma1,m}$. Note that this canonical transformation includes all modes $m$ from the first canonical transformation. Details of the second canonical transformation are given in Appendix B and illustrated in Figure 2b. We would like point out that in the full lattice, additional $4^{th}$ order interactions occur, such as ring-exchange terms, spin chirality terms [16] as well as additional $4^{th}$ order contribution to the Heisenberg and biquadratic exchange interactions. Since we restrict ourselves to a two site model, such processes are not taken into account in our calculations.

Hamiltonian Eq. (29), can be written in terms of spin-one (S=1) operators as described before in [34, 35]. In general, S=1 operators can be defined using many-electron operators [36]. Here, we define their projection onto local spin states $|S, M_S\rangle$

$$|S, M_s\rangle_i = \left\{ |1,1\rangle_i, |1,0\rangle_i, |1,-1\rangle_i \right\}. \tag{31}$$

Then, we can write the spin-one states in terms of single electron states using suitable Clebsh-Gordan coefficients

$$|1,1\rangle_i = |\uparrow,\uparrow\rangle_i, \quad |1,0\rangle_i = \frac{1}{\sqrt{2}}(|\uparrow,\downarrow\rangle_i + |\downarrow,\uparrow\rangle_i), \quad |1,-1\rangle_i = |\downarrow,\downarrow\rangle_i, \tag{32}$$

where $|\sigma_a, \sigma'_b\rangle_i = \hat{c}_{ib\sigma'}^{\dagger} \hat{c}_{ia\sigma}^{\dagger} |0\rangle$. Using the relation [36]

$$\hat{S}_i^q |S, M_S\rangle_i = \sqrt{S(S+1)} C_{SM_S,1q}^{SM_S+q} |S, M_S+q\rangle_i, \tag{33}$$

one can write $\hat{S}^q$ in terms of single electron operators (index $i$ is omitted for brevity), which yields

$$\hat{S}^{+1} = -\frac{1}{\sqrt{2}} \sum_{\alpha \neq \beta} \hat{c}_{\alpha\uparrow}^{\dagger} \hat{c}_{\alpha\downarrow} (\hat{n}_{\beta\uparrow} \hat{h}_{\beta\downarrow} + \hat{h}_{\beta\uparrow} \hat{n}_{\beta\downarrow}), \tag{34}$$

$$\hat{S}^{-1} = \frac{1}{\sqrt{2}} \sum_{\alpha \neq \beta} \hat{c}_{\alpha\downarrow}^{\dagger} \hat{c}_{\alpha\uparrow} (\hat{n}_{\beta\uparrow} \hat{h}_{\beta\downarrow} + \hat{h}_{\beta\uparrow} \hat{n}_{\beta\downarrow}), \tag{35}$$

$$\hat{S}^0 = \hat{n}_{a\uparrow} \hat{h}_{a\downarrow} \hat{n}_{b\uparrow} \hat{h}_{b\downarrow} - \hat{h}_{a\uparrow} \hat{n}_{a\downarrow} \hat{h}_{b\uparrow} \hat{n}_{b\downarrow}. \tag{36}$$

Using the definition $\hat{S}^{\pm 1} = \mp \frac{1}{\sqrt{2}}(\hat{S}^x \pm i\hat{S}^y)$ for the spin-one spin flip terms [36], one can compute the product $\vec{S}_i \cdot \vec{S}_j$ as well as $(\vec{S}_i \cdot \vec{S}_j)^2$ in terms of single electron operators and identify

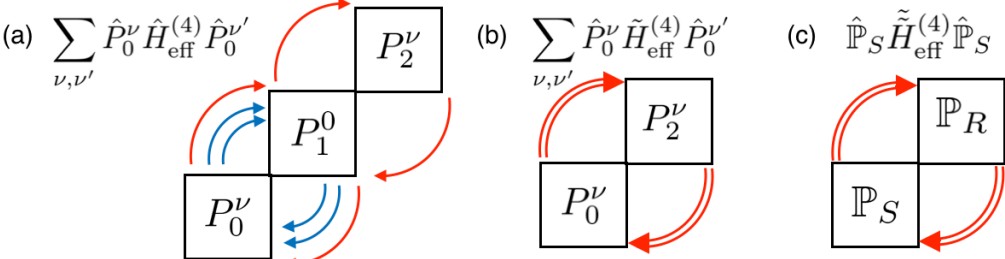

Figure 2: (Color online) Diagrams illustrating the fourth order hopping process of (a) the first canonical transformation via the $\hat{P}_0^\nu$ sector in blue and via the $\hat{P}_2^\nu$ sector in red. (b) The second canonical transformation via the $\hat{P}_2^\nu$ sector and (c) the third canonical transformation between the spin one sector $\mathbb{P}_S$ and a the non spin-one sector $\mathbb{P}_R$ from the $\hat{P}_0^\nu$ sector. Red and blue arrows represent first order hopping processes in (a), double arrows represent second order hopping processes.

them with the terms of Eq. (29). This procedure leads to an effective Hamiltonain written in terms of $S=1$, $\hat{R}_{ij}^1$ and $\hat{R}_{ij}^2$ operators, see Appendix D. Subsequently, by time averaging $\bar{H}_{\text{eff}}^{d=0}=\frac{1}{T}\int_0^T \hat{H}_{\text{eff}}^{d=0}(t)dt$, we obtain an effective time-independent Hamiltonian:

$$\bar{H}_{\text{eff}}^{d=0}= \sum_{<i,j>} \left\{ K_1(\mathcal{E},\omega)\left(\vec{S}_i\cdot\vec{S}_j + \hat{R}_{ij}^1\right) + K_2(\mathcal{E},\omega)\left(\vec{S}_i\cdot\vec{S}_j + \hat{R}_{ij}^1\right)^2 + K_3(\mathcal{E},\omega)\left((\vec{S}_i\cdot\vec{S}_j)^2 + \hat{R}_{ij}^2\right) \right\}. \quad (37)$$

$K_1(\mathcal{E},\omega)$ corresponds to the exchange $J_{\text{ex}}(\mathcal{E},\omega)$ up to second order in the hopping. $K_2(\mathcal{E},\omega)$ gives a fourth order contribution to $J_{\text{ex}}(\mathcal{E},\omega)$ as well as the biquadratic exchange $B_{\text{ex}}(\mathcal{E},\omega)$. $K_3(\mathcal{E},\omega)$ gives a contribution directly to $B_{\text{ex}}(\mathcal{E},\omega)$.

The remaining terms $\hat{R}_{ij}^1$ and $\hat{R}_{ij}^2$ describe orbital resolved spin dynamics that strictly go beyond a spin-one model. Their expression in terms of fermionic operators can be found in Appendix D. To arrive at an effective spin-one model only, we perform a third time-dependent canonical transformation between the spin one sector $\hat{\mathbb{P}}_S$ and the $S\neq 1$ sector $\hat{\mathbb{P}}_R$ from the $\hat{P}_0^\nu$ sector. In this process, ilustrated in Figure 2c, $\hat{\mathbb{P}}_R$ is taken as a high energy sector. Details of the calculations can be found in Appendix D. Eventually, we obtain an effective spin-one model $\hat{H}_{\text{ex}}=J_{\text{ex}}\vec{S}_i\cdot\vec{S}_j + B_{\text{ex}}(\vec{S}_i\cdot\vec{S}_j)^2$, where

$$J_{\text{ex}}(\mathcal{E},\omega)= \sum_{m=-\infty}^{+\infty}\left\{ \frac{t_0^2 J_m^2(\mathcal{E})}{U+J_H+m\omega} \right.$$
$$\left. -2t_0^4\sum_{k+l+m+n=0}J_k(\mathcal{E})J_l(\mathcal{E})J_m(\mathcal{E})J_n(\mathcal{E})(-1)^k\left[(-1)^m+(-1)^n\right]C_{01}^{00,k}C_{10}^{00,l}C_{01}^{00,m} \right\}, \quad (38)$$

where $J_m$ is a Bessel function of order $m$. The first term of Eq. (38) corresponds to $K_1(\mathcal{E},\omega)$ and the second term is a contribution from $K_2(\mathcal{E},\omega)$.

We now would like to compare the behavior of the second order $J_{\text{ex}}(\mathcal{E},\omega)$ in single and two-orbital systems. $J_{\text{ex}}(\mathcal{E},\omega)$ for single-orbital systems reads

$$J_{\text{ex}}^{\text{single}}(\mathcal{E},\omega) = \sum_{m=-\infty}^{+\infty} \frac{2t_0^2 J_m^2(\mathcal{E})}{U+m\omega}. \quad (39)$$

We can see that $J_{\text{ex}}(\mathcal{E},\omega)$ in the two-orbital model, Eq. (38), has an additional factor $1/2$ as compared to $J_{\text{ex}}^{\text{single}}$. This is due to the inter-orbital hopping $t_{\alpha\beta}=0$ which changes the

prefactor of $J_{\text{ex}}(\mathcal{E},\omega)$ as compared to the single-orbital case. For $t_{\alpha\beta}=t_{\alpha\alpha}=t_0$, the second order contribution of the single and two-orbital model would have the same prefactor. However, the relative modification of the exchange $\Delta J_{\text{ex}}(\mathcal{E},\omega)/J_{\text{ex}}(\mathcal{E},\omega)$ is the same and therefore, orbital dynamics does not change the control of $J_{\text{ex}}(\mathcal{E},\omega)$.

The biquadratic exchange interaction can be written as a sum of six contributions:

$$B_{\text{ex}}(\mathcal{E},\omega)=B_{\text{ex}}^{[1]}(\hat{P}_0) + B_{\text{ex}}^{[1]}(\hat{P}_2^0) + B_{\text{ex}}^{[1]}(\hat{P}_2^1) + B_{\text{ex}}^{[2]}(\hat{P}_2^0) + B_{\text{ex}}^{[2]}(\hat{P}_2^1) + B_{\text{ex}}^{[3]}(\hat{P}_0), \tag{40}$$

where

$$B_{\text{ex}}^{[1]}(\hat{P}_0) = 2\sum_{k+l+m+n=0} A_{klmn}^1(\mathcal{E})C_{01}^{00,k}C_{10}^{00,l}C_{01}^{00,m}, \tag{41}$$

$$B_{\text{ex}}^{[1]}(\hat{P}_2^0) = \sum_{k+l+m+n=0} A_{klmn}^2(\mathcal{E})C_{01}^{00,k}C_{12}^{00,l}(C_{21}^{00,m} - 3C_{10}^{00,m}), \tag{42}$$

$$B_{\text{ex}}^{[1]}(\hat{P}_2^1) = \frac{1}{2}\sum_{k+l+m+n=0} A_{klmn}^1(\mathcal{E})C_{01}^{00,k}C_{12}^{01,l}(C_{21}^{10,m} - 3C_{10}^{00,m}), \tag{43}$$

$$B_{\text{ex}}^{[2]}(\hat{P}_2^0) = \sum_{k+l+m+n=0} A_{klmn}^2(\mathcal{E})C_{02}^{00,k+l}(C_{01}^{00,k} - C_{12}^{00,k})(C_{21}^{00,m} - C_{10}^{00,m}), \tag{44}$$

$$B_{\text{ex}}^{[2]}(\hat{P}_2^1) = \frac{1}{2}\sum_{k+l+m+n=0} A_{klmn}^1(\mathcal{E})C_{02}^{01,k+l}(C_{01}^{00,k} - C_{12}^{01,k})(C_{21}^{10,m} - C_{10}^{00,m}), \tag{45}$$

$$B_{\text{ex}}^{[3]}(\hat{P}_0) = -\sum_{k+l+m+n=0} A_{klmn}^1(\mathcal{E})\frac{(C_{01}^{00,k}-C_{10}^{00,l})(C_{01}^{00,m}-C_{10}^{00,n})}{4(4J_H + (k+l)\omega)}, \tag{46}$$

with

$$A_{klmn}^1(\mathcal{E}) = t_0^4(-1)^k\big[(-1)^m+(-1)^n\big]J_k(\mathcal{E})J_l(\mathcal{E})J_m(\mathcal{E})J_n(\mathcal{E}), \tag{47}$$

$$A_{klmn}^2(\mathcal{E}) = t_0^4(-1)^{k+l}J_k(\mathcal{E})J_l(\mathcal{E})J_m(\mathcal{E})J_n(\mathcal{E}). \tag{48}$$

We used the notation $B_{\text{ex}}^{[i]}(\hat{P}_d^{\nu})$ to denote the first, second and third canonical transformation via the $\hat{P}_d^{\nu}$ sector, for $i=1,2,3$ repsectively. Since the energy approximation Eq. (24) leads to a same energy for both $\hat{P}_0^0$ and $\hat{P}_0^1$, we group the biquadratic paths via these two sectors into one path via the $\hat{P}_0$ sector, $B_{\text{ex}}^{[i]}(\hat{P}_0)=\sum_{\nu}B_{\text{ex}}^{[i]}(\hat{P}_0^{\nu})$, where $\nu=0,1$. In deriving Eqs. (41-46), one obtains factors which contain $k$, $l$, $m$ and $n$ indices, see Eqs. (47) and (48), these directly arise from the canonical transformation and come from Bessel functions $J_{-m}$ which are symmetric for even $m$ but anti-symmetric for odd $m$.

Figure 3 shows the behavior of the Heisenberg exchange and the biquadratic exchange interaction in the two-orbital model as a function of the driving strength $\mathcal{E}$. The upper panel of Figure 3a shows the typical behavior of $J_{\text{ex}}(\mathcal{E})$ while the lower panel shows behavior of $B_{\text{ex}}(\mathcal{E})$ for frequencies $\omega=9$, 18 and 25. We observe that $J_{\text{ex}}(\mathcal{E},\omega)$ can be controlled with the strength $\mathcal{E}$ and frequency $\omega$ of the electric field similarly as found in [12] for the single-orbital system i.e. $J_{\text{ex}}(\mathcal{E})$ can be reduced for frequencies above the Mott gap $U + J_H$, enhanced for frequencies below the gap and reversed for stronger driving field $\mathcal{E}$. The major contribution to $J_{\text{ex}}(\mathcal{E})$ comes from the second order contribution in the hopping.

Figure 3b shows the contributions of the different biquadratic paths $B_{\text{ex}}^{[i]}(\hat{P}_d^{\nu})$ as a function of the driving strength $\mathcal{E}$. The top panel shows $B_{\text{ex}}^{[1]}(\hat{P}_0^{\nu})$ in red and $B_{\text{ex}}^{[2]}(\hat{P}_2^0)$ in blue which are

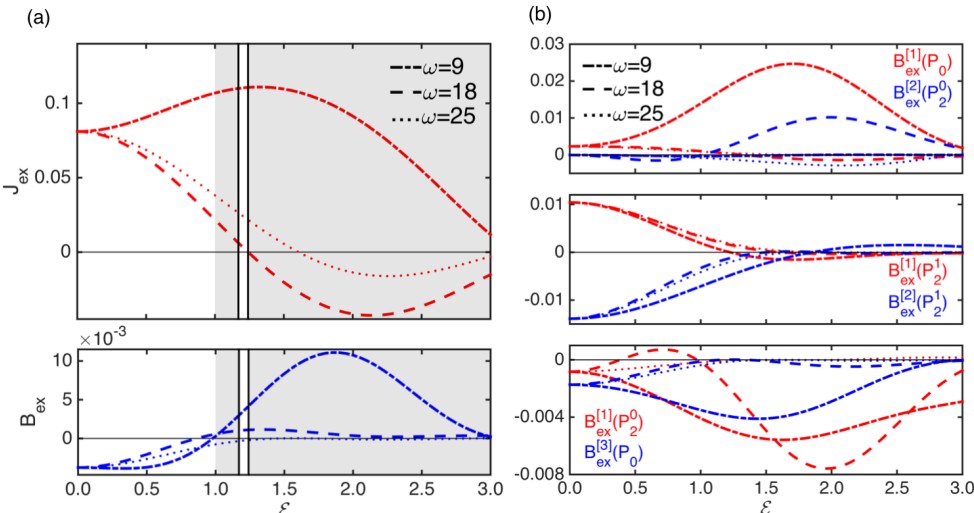

Figure 3: (Color online) (a) $J_{\text{ex}}(\mathcal{E}, \omega)$ in the upper panel and $B_{\text{ex}}(\mathcal{E}, \omega)$ in the lower panel as a function of $\mathcal{E}$, the grey area represents regime for which both $J_{\text{ex}}(\mathcal{E}, \omega)$ and $B_{\text{ex}}(\mathcal{E}, \omega)$ are positive for $\omega=9$ and rectangular box represents the regime for which $|J_{ex}(\mathcal{E}, \omega)| \sim |B_{\text{ex}}(\mathcal{E}, \omega)|$ for $\omega=18$. (b) Biquadratic exchange paths $B_{\text{ex}}^{[i]}(\mathcal{E})$ as a function of $\mathcal{E}$, where $i=1, 2, 3$ indicates the canonical transformation order set in Section 3.1. Results are computed for $\omega=9$ (dash-dot line), $\omega=18$ (dashed line) and $\omega=25$ (dots), the frequency $\omega$ is expressed in units of the hopping $t_0$. Parameters for the Figure: $U=10/t_0$, $J_H=2/t_0$.

the strongest contributions to the biquadratic exchange. The middle panel displays $B_{\text{ex}}^{[1]}(\hat{P}_2^1)$ in red and $B_{\text{ex}}^{[2]}(\hat{P}_2^1)$ in blue. On the bottom panel, we plotted the weakest contributions to the biquadratic exchange: $B_{\text{ex}}^{[1]}(\hat{P}_2^0)$ in red and $B_{\text{ex}}^{[2]}(\hat{P}_0)$ in blue. By summing up all the $B_{\text{ex}}^{[i]}(\hat{P}_d^\gamma)$ paths the biquadratic exchange $B_{\text{ex}}(\mathcal{E})$ is obtained as shown in the bottom panel of Figure 3a. In equilibrium, $B_{\text{ex}}(\mathcal{E}=0)<0$ favors a collinear alignment of spins in the classical limit and $|B_{\text{ex}}(\mathcal{E}=0)|$ is weak as compared to $|J_{\text{ex}}(\mathcal{E}=0)|$.

We observe that analogous to $J_{\text{ex}}(\mathcal{E}, \omega)$, also $B_{\text{ex}}(\mathcal{E}, \omega)$ can be controlled by the electric field strength and frequency. In the regime of low driving field strength $\mathcal{E} \ll 1$, $|B_{\text{ex}}(\mathcal{E}, \omega)|$ is reduced for frequencies above the Mott gap, $\omega=18$ and 25, and enhanced for the frequency below the gap, shown here for $\omega=9$. The enhancement of $|B_{\text{ex}}(\mathcal{E})|$ can be understood from Figure 3b where $|B_{\text{ex}}^{[1]}(\hat{P}_2^0)|$ and $|B_{\text{ex}}^{[3]}(\hat{P}_0)|$ are both enhanced at low driving field. In addition, the sum of $B_{\text{ex}}^{[1]}(\hat{P}_2^1)$ and $B_{\text{ex}}^{[2]}(\hat{P}_2^1)$ gives a reduction of $B_{\text{ex}}$. Eventualy, the sum these four contributions dominates over the large enhancement of the $B_{\text{ex}}^{[1]}(\hat{P}_0)$ contribution leading to an enhancement of $|B_{\text{ex}}(\mathcal{E})|$. The physical mechanism behind the increase/reduction of $|B_{\text{ex}}(\mathcal{E}, \omega)|$ for low driving field strength can be explained as follows: the virtual hopping to $m\omega$ high energy states is enhanced or reduced as compare to equilibrium. This leads to an increase, decrease or change of sign of the $C_{dd'}^{\nu\nu', m}$ products in different biquadratic paths $B_{\text{ex}}^{[i]}(\hat{P}_d^\gamma)$, Eqs. (41-46). Summing all the biquadratic paths, the total $|B_{\text{ex}}(\mathcal{E}, \omega)|$ is enhanced or reduced as compare to its equilibrium value.

For larger driving field strength $\mathcal{E} \gtrsim 1$, the reduction of photo assisted hopping as well as the oscillatory origin of the Bessel function can lead to a change of sign of $B_{\text{ex}}(\mathcal{E}, \omega)$. Interestingly, for frequency $\omega=9$, we identify a regime for which both $J_{\text{ex}}(\mathcal{E})$ and $B_{\text{ex}}(\mathcal{E})$ are positive for a range of driving field strength $\mathcal{E}>1$, this regime is diplayed with a gray area in Figure 3a. At $\omega=18$, the rectangle box in Figure 3a shows a regime for which $J_{\text{ex}}(\mathcal{E}) \sim B_{\text{ex}}(\mathcal{E})>0$. Within this regime, both $J_{\text{ex}}(\mathcal{E})$ and $B_{\text{ex}}(\mathcal{E})$ are positive which leads to a competition between the exchange

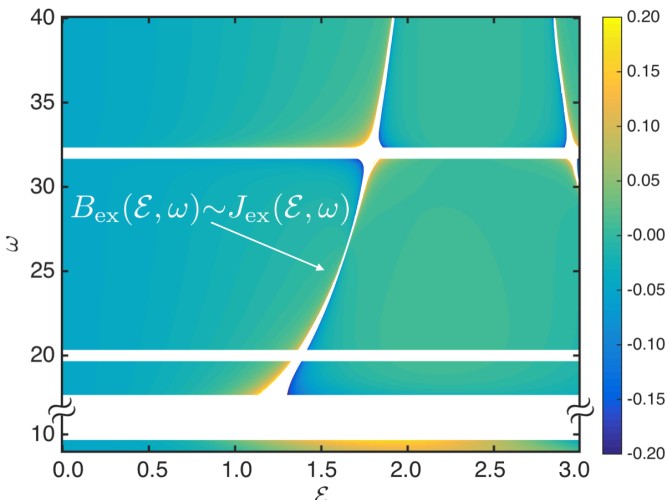

Figure 4: (Color online) Value of the exchange ratio $B_{\mathrm{ex}}(\mathcal{E}, \omega)/J_{\mathrm{ex}}(\mathcal{E}, \omega)$ for frequencies $9 \leq \omega \leq 40$ and driving amplitude $\mathcal{E}$ up to 3. White lines at $\omega = 10, 12, 16, 20$ and 32 red are centered around frequencies at chich the canonical transformation diverges. The frequency range $11 \leq \omega \leq 17$ is not displayed in the figure since the bi-quadratic formula is not accurate in this range. White curved areas correspond to points for which $B_{\mathrm{ex}}(\mathcal{E}, \omega) \sim J_{\mathrm{ex}}(\mathcal{E}, \omega)$. Parameters: $U/t_0 = 10$, $J_H/t_0 = 2$. For clarity, $B_{\mathrm{ex}}(\mathcal{E}, \omega)/J_{\mathrm{ex}}(\mathcal{E}, \omega)$ is restricted to $-0.23$ to $0.23$.

interactions. We note that our results suggest that in principle it is possible to change the ratio of $J_{\mathrm{ex}}(\mathcal{E}) \sim B_{\mathrm{ex}}(\mathcal{E})$ over a large range, where the equilibrium phase diagram in 1D shows several distinct quantum phases. It will be very interesting to study the feasibility of dynamical transitions between such phases in future work. Analogously, it might be very interesting to study the emergence of non-collinear order in classical spin systems by perturbation of the ratio $J_{\mathrm{ex}}(\mathcal{E}) \sim B_{\mathrm{ex}}(\mathcal{E})$. For resonant photo-excitation, this problem has been studied recently and it was indeed found that the non-collinear phase can emerge [22].

Next, we study the possible enhancement of $B_{\mathrm{ex}}(\mathcal{E}, \omega)/J_{\mathrm{ex}}(\mathcal{E}, \omega)$ for a wider range of freqeuncies ($9 \leq \omega \leq 40$) where $\omega = 40$ is larger than the highest energy of the undriven system $E_2^0 - E_0^0$. The result is shown in Figure 4 as a color map as a function of $\mathcal{E}, \omega$. Positive values of the ratio are shown in yellow and negative values are shown in blue. Below the Mott gap ($\omega = 12$), accurate results can only be obtained in a frequency range $9 \leq \omega \leq 9.5$. Below and above this range until $\omega \simeq 17$, the energy approximation Eq. (24) breaks down and orbital resolved spin dynamics [34] is required to have an accurate description of the exchange interactions. Therefore, the frequency range $10 \leq \omega \leq 17$ is not shown. Figure 4 clearly demonstrates that the exchange ratio can be enhanced as well as reduced depending on $\omega$ and $\mathcal{E}$. The parameters for which $B_{\mathrm{ex}}(\mathcal{E}, \omega)/J_{\mathrm{ex}}(\mathcal{E}, \omega)$ is strongly enhanced correspond to three types of situations:

- Frequencies for which $\omega = E_d^\gamma - E_{d'}^{\gamma'} + m\omega$, this corresponds to white lines at $\omega = 10, 12, 16, 20$ and 32. At these frequencies, $B_{\mathrm{ex}}(\mathcal{E}, \omega)$ diverges, such that the canonical transformation breaks down. Note that $\omega = 32$ is the frequency that separates spin states from the doubly ionized state, such that a coupling appears close to this frequency. This coupling leads to charge dynamics and therefore, the spin-one model is not accurate in this region. This coupling to charge states is studied in detail in the next section.

- Field parameters $\omega$ and $\mathcal{E}$ for which $B_{\mathrm{ex}}(\mathcal{E}, \omega) \sim J_{\mathrm{ex}}(\mathcal{E}, \omega)$ are indicated in white curved areas. For these regime, the exchange ratio $|B_{\mathrm{ex}}(\mathcal{E}, \omega)/J_{\mathrm{ex}}(\mathcal{E}, \omega)|$ is enhanced since $J_{\mathrm{ex}}(\mathcal{E}, \omega) \simeq 0$. This leads to a regime where $B_{\mathrm{ex}}(\mathcal{E}, \omega) > J_{\mathrm{ex}}(\mathcal{E}, \omega)$ is realised however, $B_{\mathrm{ex}}(\mathcal{E}, \omega)$ itself remains

small as compared to $J_{ex}(\mathcal{E}=0)$.

• Parameters $\omega$ and $\mathcal{E}$ for which the relative sign of $B_{ex}(\mathcal{E}, \omega)/J_{ex}(\mathcal{E}, \omega)$ is changed due to the change of sign of $B_{ex}(\mathcal{E}, \omega)$ leading to a slight enhancement of $B_{ex}(\mathcal{E}, \omega)/J_{ex}(\mathcal{E}, \omega)$. This can be clearly seen for frequency $\omega \simeq 9$ at $\mathcal{E} \simeq 2$.

Summarizing, orbital degrees of freedom do not change the behavior of $J_{ex}(\mathcal{E}, \omega)$. Both sign and strength of $J_{ex}(\mathcal{E}, \omega)$ and $B_{ex}(\mathcal{E}, \omega)$ as well as their relative sign can be controlled by driving, while the regime for which $B_{ex}(\mathcal{E}, \omega) \sim J_{ex}(\mathcal{E}, \omega)$ is reached only for $J_{ex}(\mathcal{E}, \omega) \ll J_{ex}(\mathcal{E}=0)$.

### 3.2 Beyond the spin-one model

Besides the additional term $B_{ex}$, the inclusion of orbital degrees of freedom also gives rise to qualitatively new effects that go beyond a description in terms of a spin model alone. In particular, under driving there can be coupling to the doubly ionized charge sector ($d=2$), which is irrelevant in equilibrium due to the large energy difference between the sectors. Under non-equilibrium conditions, the spin charge coupling reads

$$\hat{H}_{sc}^{(n)}(t) = \sum_n \sum_{\nu=0,1} \hat{P}_0^{\nu} \hat{H}_{eff}^{(n)}(t) \hat{P}_2^{\nu'} + h.c. \tag{49}$$

We now study the regime for which $\hat{P}_0^{\nu}$ and $\hat{P}_2^{\nu'}$ from different $m$ sectors overlap. This overlap appears for frequencies $\omega$ close to $E_0^{\nu} = E_2^{\nu} + m\omega$. Although this seems a resonant condition, a direct optical transition is not possible since two hoppings are required to go from the $\hat{P}_0^{\nu}$ sector to the $\hat{P}_2^{\nu'}$ sector. Equation (49) can be divided into contributions $\sum_m \hat{H}_{sc,\Delta m=0}^{(n)}$ and $\sum_m \hat{H}_{sc,\Delta m \neq 0}^{(n)}$. The first contribution represents the coupling within one Fourier sector $m$. This contribution remains weak since $\hat{P}_0^{\nu}$ and $\hat{P}_2^{\nu'}$ states are strongly gapped when they belong to the same Fourier sector. We therefore focus on the second term that allows coupling between the spin sector $\hat{P}_0^{\nu}$ and the charge sector $\hat{P}_2^{\nu'}$ with different $m$. For small $\mathcal{E}$, the leading contribution to Eq. (49) arises from $n=2$ and $\Delta m = \pm 1$. Here we restrict to the coupling between between $\hat{P}_0^{\nu}$ from $m=0$ and $\hat{P}_2^{\nu'}$ from $m=-1$ sector. This yields

$$\hat{H}_{sc,|\Delta m|=1}^{(2)}(t) = \frac{1}{2} \sum_k \left( \frac{1}{3U - 5J_H - k\omega} - \frac{1}{U + J_H - k\omega} \right)$$
$$\times \left[ \sum_{\nu,\nu'} \hat{P}_0^{\nu} \hat{T}_k^{-1} \hat{P}_1^0 \hat{T}_{1-k}^{-1} \hat{P}_2^{\nu'} e^{-i\omega t} \right] + h.c. \sim \mathcal{E}. \tag{50}$$

To illustrate the spin-charge coupling, we restrict ourselves to the space $\nu'=0$. These are two states that have all electrons either on site $i$ or on site $j$. The full expression of $\hat{H}_{sc,|\Delta m|=1}^{(2)}(t)$ in terms of fermionic operators is given in Appendix E.

To show the effect of this coupling, we compute the low energy spectrum for driving frequencies $\omega = \omega_0 + \delta\omega$, where $\omega_0 = |E_0^{\nu} - E_2^0|$. The spectrum is shown in Figure 5 for $\delta\omega = 0.5$ in the two-site system. In this case, the lowest energy state is a singlet state that couples to the two charge states of $\hat{P}_2^0$. The latter are degenerate up to $t_0^4$ since four hoppings are needed in order to transfer the four electrons from one atom to the other one. Black lines, from top to bottom, show the quintet state ($S=2$) and the triplet state ($S=1$) that are not involved in the spin-charge coupling. The black dashed line shows the behavior of the spin state which is a singlet state from sector $m=0$. The dotted lines show the behavior of the charge states from $m=-1$. The thick red and blue lines show the spin and charge states, respectively and are obtained by diagonalizing the full $\hat{H}_{eff}^{(2)}$ that contains the spin-charge coupling terms, see Appendix E. The eigen-energies show an avoided crossing, which reveals a hybridization between the spin and charge states. For driving frequencies far away from $\omega_0$, the spin-charge

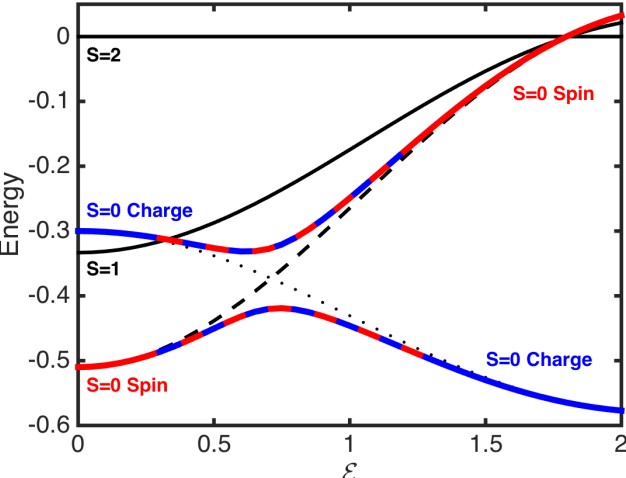

Figure 5: (Color online) Low energy spectrum as a function of the driving amplitude $\mathcal{E}$. The spectrum is restricted to the spin states of the $m=0$ sector and the highest excited state of the sector $m=-1$. Upper and lower thin black lines represent the quintet state and the triplet state. The thin black dots and the dashed line represent the charge multiplet and the singlet without spin-charge coupling, respectively. Blue to red thick line represents the charge ($\mathcal{E}<0.5$) to spin ($\mathcal{E}>1$) state and the red to blue thick line represents the spin ($\mathcal{E}<0.5$) to charge ($\mathcal{E}>1$) state. Parameters: $U/t_0=10$, $J_{\mathrm{H}}/t_0=2$ and $\omega=\omega_0+0.5$.

coupling terms $\hat{H}^{(2)}_{sc,|\Delta m|=1}(t)$ remain small and for these frequencies, the spin and charge states are gapped. Therefore, the hybridization is negligible and we recover the regime for which the effective spin model is valid. However, in the regime $\omega\sim\omega_0$, the hybridization between the spin and charge states cannot be neglected and the exchange interaction formula obtained in the previous section are no longer accurate.

To sketch the hybridization process, let us take the equilibrium ground state state of the system which is the singlet state (spin state). After switching on the electric field and by slowly changing the field amplitude ($\partial_t E_0/E_0\ll\omega$), one anticipates that the system starts in a pure spin state and, approaching the avoided-crossing regime, charge states are mixed to the state. For strong $\mathcal{E}$, the spin state becomes a pure charge state.

In the literature, a time-dependent traverse of an avoided crossing is widely studied. The basic example is the Landau-Zener (LZ) effect [38, 39]. Also condensed matter systems can exhibit LZ physics. For instance, the Zener breakdown has been studied [40, 41] in semiconductors and more recently in Mott insulators [42]. Nonetheless, distinct from these LZ effects which involve changes of the electrical conductivity, here we report coherent transfer of spin to charge degrees of freedom that keep the system in an insulating regime.

Summarizing, orbital dynamics gives access to charge states that are not accessible in equilibrium. The spin-charge coupling offers the possibility to induce coherent charge dynamics in the system. This phenomenon appears in the non-equilibrium low-energy spectrum as an avoided crossing. We stress that it is quite distinct from spin-orbit coupling since here we have an interplay with Coulomb and Hund interaction with the driving field. The spin-charge coupling is not present in single band systems, and we expect it to be universal for multi-orbital systems. Indeed, since multi-orbital systems offer the possibility of having multiple excited states (multiple doublons), multi-doublon excitation should be possible for multi-orbital systems in general. Note that here we did not study $\hat{P}^1_2$ states, they are nonetheless very interesting since they have a lower energy (at and below gap energy $U+J_H$) and therefore are reachable

with lower frequencies $\omega$.

# 4  Time-dependent numerical simulations

In the previous section, we showed that the generalized canonical transformation can capture a qualitatively new phenomenon that couples the spin and doubly ionized charge sector. To further support this finding, we perform an exact time propagation of a cluster of two sites described by the Mott-Hubbard Hamiltonian. We focus our attention on the coherent transfer of spin to charge degrees of freedom. In order to describe the charge dynamics, we define pseudo-spin one operators $\hat{\mathcal{T}}^0$ that are composed of Anderson pseudo-spin 1/2 operators $\hat{\tau}^+_{i\alpha}=\hat{c}^\dagger_{i\alpha\uparrow}\hat{c}^\dagger_{i\alpha\downarrow}$ ($\hat{\tau}^-_{i\alpha}=\hat{c}_{i\alpha\downarrow}\hat{c}_{i\alpha\uparrow}$) [43, 44]. The construction of $\hat{\mathcal{T}}^0$ is inspired by the expression of the spin-one operator $\hat{S}^0$. By using a similar procedure with $\hat{\mathcal{T}}^0$ and pseudo-spin 1/2 operators, we obtain

$$\hat{\mathcal{T}}^0 = \hat{\tau}^+_a \hat{\tau}^-_a \hat{\tau}^+_b \hat{\tau}^-_b - \hat{\tau}^-_a \hat{\tau}^+_a \hat{\tau}^-_b \hat{\tau}^+_b = \hat{n}_{a\uparrow}\hat{n}_{a\downarrow}\hat{n}_{b\uparrow}\hat{n}_{b\downarrow} + \hat{h}_{a\uparrow}\hat{h}_{a\downarrow}\hat{h}_{b\uparrow}\hat{h}_{b\downarrow}, \tag{51}$$

that characterizes fully occupied and completely empty sites from the $\hat{P}^0_2$ sector. Operators $\hat{\mathcal{T}}^{+1}$ and $\hat{\mathcal{T}}^{-1}$ can be defined analogously, see Appendix C.

To solve the time-dependent Schrödinger equation, we use a second order commutator-free approximation of the time-propagator [45] and we compute the time-dependent wavefunction $|\Psi(t)\rangle$ and evaluate observables as $\langle\hat{O}\rangle=\langle\Psi(t)|\hat{O}|\Psi(t)\rangle$. We focus on three different observables: First, the spin correlation $\langle\vec{S}_i\cdot\vec{S}_j\rangle$ to show the spin dynamics during the laser pulse. Second, we characterize the charge states $\hat{P}^0_2$ with $\langle\hat{\mathcal{T}}^0_i\hat{\mathcal{T}}^0_j\rangle$. Finally, to probe the states that possess one doublon $d=1$, we evaluate $\langle\hat{N}_{d=1}\rangle$, where $\hat{N}_{d=1}=\hat{P}^0_1\hat{d}\hat{P}^0_1$.

Figure 5 shows simulated spin-charge dynamics for an electric field $E(t)=E_0\cos(\omega t)\times\exp\bigl(-(t-t_*)^2/\tau^2\bigr)$, where $E_0$ is the amplitude of the field, $t_*$ is the time at which $E(t)$ peaks and $\tau$ is the pulse width.

We choose a Gaussian envelope with $\tau=4000\pi/\omega$, $\omega=\omega_0+\delta\omega$, with $\delta\omega=0.5$, such that $\tau\times\omega_{sc}\gg1$ where $\omega_{sc}\simeq0.1$ is the energy splitting of the avoided crossing (see Figure 5). It has been shown that the effective Hamiltonian picture can break down for long-time dynamics in the thermodynamic limit, because the system heats up to infinite temperature [46]. Here we restrict ourselves to a two-site system to mimic the dynamics of a large system at relatively short timescales. However, for generic systems it is shown that heating can occur at short timescale since the adiabatic limit of Floquet does not exists [47]. Nevertheless, here the use of Floquet restricts to the derivation of an effective Hamiltonian which gives a qualitative picture of the avoided-crossing. We confirm the reversibility of the spin-charge coupling phenomenon within the two-site system with the time-dependent numerical simulations displayed in Figure 6.

Figure 6a, c show the plot of the charge and spin observables respectively, for different driving strength $\mathcal{E}$ from 0.1 to 1.5. The time-dependent electric field is represented in light blue and the results are computed for $U/t_0=10$ and $J_H/t_0=2$.

Figure 6c shows $\langle\vec{S}_i\cdot\vec{S}_j\rangle$ for different $\mathcal{E}$. In equilibrium and for small $\mathcal{E}$, $\langle\vec{S}_i\cdot\vec{S}_j\rangle\simeq-1.9$, which slightly deviates from the pure spin case ($\langle\vec{S}_i\cdot\vec{S}_j\rangle=-2$) due to hybridization with $\hat{P}^\gamma_0$, $\hat{P}^0_1$ and $\hat{P}^0_2$ sectors. Figure 6c shows that, with increasing $\mathcal{E}$ the state has less spin characteristics. Moreover, it is observed in Figure 6a that with increasing $\mathcal{E}$, the state has more charge characteristics $\langle\hat{\mathcal{T}}^0_i\hat{\mathcal{T}}^0_j\rangle$. In addition, after the laser pulse, both charge excitation and the spin correlations return to their initial value, demonstrating that the coupling is reversible.

Further, we confirm that for frequencies away from $\omega_0$ the spin-charge coupling dynamics is not present. This is shown in Figure 6a,c where the charge and spin dynamics are plotted in red lines for $\delta\omega=3$. In this case, no enhancement is observed. Similarly, in Figure 6c, the spin correlations are not diminished. Moreover, we show the single doublon states dynamics

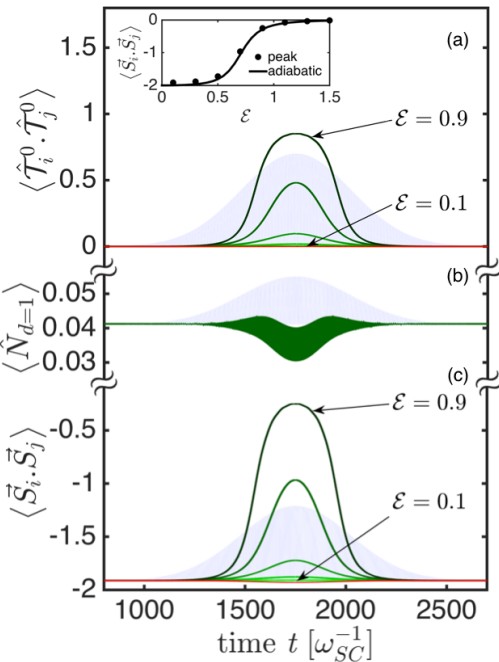

Figure 6: (Color online) (a) and (c) Spin $\left\langle \vec{S}_i \cdot \vec{S}_j \right\rangle$ and charge $\left\langle \hat{\mathcal{T}}_i^0 \hat{\mathcal{T}}_j^0 \right\rangle$ dynamics as a function of time in orange and green, respectively. The dynamics is computed for driving amplitudes $\mathcal{E}$ from 0.1 to 0.9 with steps $\Delta\mathcal{E}$=0.2 represented by different color shades. (b) Single doublon number $\left\langle \hat{N}_{d=1} \right\rangle$ in blue, for $\mathcal{E}$=0.7. The amplitude of the electric field envelop is shown in Figure (a),(b) and (c) by a light blue Gaussian, each laser pulse contains 4000 cycles. Parameters $U/t_0$=10 and $J_{\mathrm{H}}/t_0$=2, for an electric field frequency $\omega=\omega_0+0.5$. The inset shows a comparison between the time evolution (dots) and the analytical calculation in the adiabatic limit for the field envelope (solid line). Red lines in (a) and (c) represent the spin and charge dynamics for frequency $\omega=\omega_0+3$ and $\mathcal{E}$=0.7, away from the hybridization.

$\left\langle \hat{N}_{d=1} \right\rangle$ in Figure 6b, for a field strength $\mathcal{E}$=0.7. We observe that the laser pulse does not trigger any positive excitation of $\hat{P}_1^0$ states. This means that enhancement of charge dynamics is not due to resonant excitation of the intermediate excited states $\hat{P}_1^0$. Interestingly, we actually observe depopulation of the $\hat{P}_1^0$ states during the laser pulse. This means that more $\hat{P}_1^0$ states are virtually excited to the doubly inonized sector than spin states excited to the $\hat{P}_1^0$ sector. Finally, the inset of Figure 6a shows values of the peak of the spin correlations $\left\langle \vec{S}_i \cdot \vec{S}_j \right\rangle_{t_{\mathrm{peak}}}$ for each $\mathcal{E}$ in dots and values of the spin correlations $\left\langle \vec{S}_i \cdot \vec{S}_j \right\rangle$ as a function of $\mathcal{E}$ is obtained from the anaytical calculation of Section 3.2. Good agreement between analytical and numerical results is found, which confirms the predictions of the analytical theory. The slight discrepancies between $\left\langle \vec{S}_i \cdot \vec{S}_j \right\rangle_{\mathcal{E}}$ and $\left\langle \vec{S}_i \cdot \vec{S}_j \right\rangle_{t_{\mathrm{peak}}}$ at zero field $\mathcal{E}$ can be reduced by taking into account higher order terms in the effective Hamiltonain as well as the spin-charge coupling dynamics to the full $P_2^y$ sector. In addition, we note that with careful tuning of $\delta\omega$, the reduction of $|\left\langle \vec{S}_i \cdot \vec{S}_j \right\rangle|$ as a function of $\mathcal{E}$ at small $\mathcal{E}$ can be made even stronger *e.g.* by increasing $\delta\omega$, one can move the avoided-crossing closer to $\mathcal{E}$=0 which enhances hybridization with the charge states at small $\mathcal{E}$.

# 5   Conclusion

In summary, we obtain an analytical expression for the Heisenberg $J_{ex}(\mathcal{E}, \omega)$ and biquadratic $B_{ex}(\mathcal{E}, \omega)$ exchange interaction in a periodically driven two-orbital system. We show that $J_{ex}(\mathcal{E}, \omega)$ can be controlled analogous to the single-orbital case. We find that for low driving strength, $|B_{ex}(\mathcal{E}, \omega)|$ can be reduced for frequencies above the Mott gap and enhanced for frequencies below the gap. In addition, we show that $B_{ex}(\mathcal{E}, \omega)$ can even change sign for stronger driving field srength. We demonstrate that $B_{ex}(\mathcal{E}, \omega)/J_{ex}(\mathcal{E}, \omega)$ can be controled by driving while the regime for which $B_{ex}(\mathcal{E}, \omega) \sim J_{ex}(\mathcal{E}, \omega)$ is reached only for $J_{ex}(\mathcal{E}, \omega) \ll J_{ex}(\mathcal{E} = 0)$. Moreover, a new coupling between spin and charge states is demonstrated. While this coupling is negligible in equilibrium, it can be strongly enhanced and even dominate under driving. This coupling leads to a hybridization between spin and charge states for frequencies close to the spin-charge gap. In contrast to a common charge excitation by resonant photo-absorption, the spin-charge coupling allows non-resonant and reversible coupling to charge degrees of freedom. We have furthermore confirmed these results by simulating the electron dynamics on a two-site cluster.

Natural extension of this work are numerical studies for extended systems. This could be possible for example using multiband extensions of nonequilibrium Dynamical Mean Field Theory (DMFT) [48–52]. We emphasize that, besides the possibility to induce coherent charge dynamics, the presence of the spin-charge coupling should also be visible for short pulses, enabling the excitation of doubly ionized states which could remain coherent due to the gapping with the normal Mott-Hubbard gap. This suggests interesting perspectives for enhancing electronic coherence in correlated electron systems. In this context, it will be interesting to explore the applicability of the two-orbital model to experimental spin-one systems such as $KNiF_3$ [8], which in the literature are considered as prototypical $S=1$ systems. We would like to point out that modification of the charge occupation is known to systematically influence phonon excitation. Since we did not take into account electron-phonon interactions in our model, an outlook would be to study phonon excitation induced by the spin-charge coupling. In addition, interesting prospect of this work is the two-orbitals system under arbitrary fields similarly to what is done in [53] and, since our approach with the canonical transformation can also be applied for arbitrary time-dependent fields [54]. Its extention to more exotic forms of time-dependent fields seems feasible and is left for future work. Finally, we hope that our work can find applications in cold atoms systems, where multi-orbital systems can nowadays be engineered [55, 56]. With an adiabatic ramping of the electric field strength, the fully reversible spin-to-charge conversion might be directly observed in double-well systems.

## Acknowledgements

This work was partially supported by the NWO via the Spinoza Prize, by a Rubicon and a VENI grant, by the European Research Council (ERC) Advanced Grant No. 338957 (FEMTO/NANO) and Advanced Grant No. 339813 (EXCHANGE), and the Shell-NWO/FOM-initiative "Computational sciences for energy research" of Shell and Chemical Sciences, Earth and Life Sciences, Physical Sciences, FOM and STW.

## A   Projection operators

In this section, we express the projection operators $\hat{P}_d^{\nu}$ as well as the hopping operators $\hat{T}^{\pm 1}$, $\hat{T}^0$ in terms of the electron operators $\hat{c}_{i\alpha\sigma}^{\dagger}(\hat{c}_{i\alpha\sigma})$.

$$\hat{P}_0^0 = \sum_\sigma \hat{n}_{ia\sigma}\hat{h}_{ia\bar{\sigma}}\hat{n}_{ib\sigma}\hat{h}_{ib\bar{\sigma}}\hat{h}_{ja\sigma}\hat{n}_{ja\bar{\sigma}}\hat{h}_{jb\sigma}\hat{n}_{jb\bar{\sigma}}, \tag{52}$$

$$\hat{P}_0^1 = \sum_{\alpha\neq\beta} \hat{n}_{ia\uparrow}\hat{h}_{ia\downarrow}\hat{h}_{i\beta\uparrow}\hat{n}_{i\beta\downarrow}\big(\hat{n}_{ja\uparrow}\hat{h}_{ja\downarrow}\hat{h}_{j\beta\uparrow}\hat{n}_{j\beta\downarrow} + \hat{h}_{ja\uparrow}\hat{n}_{ja\downarrow}\hat{n}_{j\beta\uparrow}\hat{h}_{j\beta\downarrow}\big), \tag{53}$$

$$\hat{P}_1^0 = \sum_{<i,j>}\sum_{\alpha\neq\beta,\sigma} \hat{n}_{ia\sigma}\hat{n}_{ia\bar{\sigma}}\hat{n}_{i\beta\sigma}\hat{h}_{i\beta\bar{\sigma}}(\hat{h}_{ja\sigma}\hat{h}_{ja\bar{\sigma}}\hat{h}_{j\beta\sigma}\hat{n}_{j\beta\bar{\sigma}} + \hat{h}_{ja\sigma}\hat{n}_{ja\bar{\sigma}}\hat{h}_{j\beta\sigma}\hat{h}_{j\beta\bar{\sigma}}), \tag{54}$$

$$\hat{P}_1^1 = \sum_{<i,j>}\sum_{\alpha\neq\beta,\sigma} \hat{n}_{ia\sigma}\hat{n}_{ia\bar{\sigma}}\hat{h}_{i\beta\sigma}\hat{h}_{i\beta\bar{\sigma}}\hat{n}_{ja\sigma}\hat{h}_{ja\bar{\sigma}}\hat{h}_{j\beta\sigma}\hat{n}_{j\beta\bar{\sigma}}, \tag{55}$$

$$\hat{P}_2^0 = \sum_{<i,j>} \hat{h}_{ia\uparrow}\hat{h}_{ia\downarrow}\hat{h}_{ib\uparrow}\hat{h}_{ib\downarrow}\hat{n}_{ja\uparrow}\hat{n}_{ja\downarrow}\hat{n}_{jb\uparrow}\hat{n}_{jb\downarrow}, \tag{56}$$

$$\hat{P}_2^1 = \sum_{\alpha\neq\beta} \hat{n}_{ia\uparrow}\hat{n}_{ia\downarrow}\hat{h}_{i\beta\uparrow}\hat{h}_{i\beta\downarrow}\big(\hat{n}_{ja\uparrow}\hat{n}_{ja\downarrow}\hat{h}_{j\beta\uparrow}\hat{h}_{j\beta\downarrow} + \hat{h}_{ja\uparrow}\hat{h}_{ja\downarrow}\hat{n}_{j\beta\uparrow}\hat{n}_{j\beta\downarrow}\big). \tag{57}$$

From Eqs. (11-13) of the main text, we can compute the following expressions for $T^{\pm 1}$ and $T^0$ in terms of single electron operators

$$\hat{T}_m^{+1}(t) = -t_0 J_m(\mathcal{E}) \sum_{\alpha\neq\beta,\sigma} \Big\{ \hat{n}_{ia\bar{\sigma}}\hat{c}_{ia\sigma}^\dagger\hat{c}_{ja\sigma}\hat{h}_{ja\bar{\sigma}}(\hat{n}_{i\beta\bar{\sigma}}\hat{h}_{j\beta\bar{\sigma}} + \hat{h}_{i\beta\bar{\sigma}}\hat{n}_{j\beta\bar{\sigma}})$$
$$+ (-1)^m \hat{n}_{ja\bar{\sigma}}\hat{c}_{ja\sigma}^\dagger\hat{c}_{ia\sigma}\hat{h}_{ia\bar{\sigma}}(\hat{n}_{i\beta\bar{\sigma}}\hat{h}_{i\beta\bar{\sigma}} + \hat{h}_{j\beta\bar{\sigma}}\hat{n}_{i\beta\bar{\sigma}})\Big\}e^{im\omega t}, \tag{58}$$

$$\hat{T}_m^{-1}(t) = -t_0 J_m(\mathcal{E}) \sum_{\alpha\neq\beta,\sigma} \Big\{ \hat{h}_{ia\bar{\sigma}}\hat{c}_{ia\sigma}^\dagger\hat{c}_{ja\sigma}\hat{n}_{ja\bar{\sigma}}(\hat{n}_{i\beta\bar{\sigma}}\hat{h}_{j\beta\bar{\sigma}} + \hat{h}_{i\beta\bar{\sigma}}\hat{n}_{j\beta\bar{\sigma}})$$
$$+ (-1)^m \hat{h}_{ja\bar{\sigma}}\hat{c}_{ja\sigma}^\dagger\hat{c}_{ia\sigma}\hat{n}_{ia\bar{\sigma}}(\hat{n}_{i\beta\bar{\sigma}}\hat{h}_{i\beta\bar{\sigma}} + \hat{h}_{j\beta\bar{\sigma}}\hat{n}_{i\beta\bar{\sigma}})\Big\}e^{im\omega t} \tag{59}$$

and

$$\hat{T}_m^0(t) = -t_0 J_m(\mathcal{E}) \sum_{\alpha\neq\beta,\sigma} \Big\{ \hat{c}_{ia\sigma}^\dagger\hat{c}_{ja\sigma}(\hat{h}_{i\beta\sigma}\hat{n}_{i\beta\bar{\sigma}}\hat{n}_{j\beta\sigma}\hat{n}_{j\beta\bar{\sigma}} + \hat{n}_{i\beta\sigma}\hat{n}_{i\beta\bar{\sigma}}\hat{h}_{j\beta\sigma}\hat{n}_{j\beta\bar{\sigma}})$$
$$+ (-1)^m \hat{c}_{ja\sigma}^\dagger\hat{c}_{ia\sigma}(\hat{h}_{j\beta\sigma}\hat{n}_{j\beta\bar{\sigma}}\hat{n}_{i\beta\sigma}\hat{n}_{i\beta\bar{\sigma}} + \hat{n}_{j\beta\sigma}\hat{n}_{j\beta\bar{\sigma}}\hat{h}_{i\beta\sigma}\hat{n}_{i\beta\bar{\sigma}})\Big\}e^{im\omega t}. \tag{60}$$

## B  Effective Hamiltonian

In this section we provide explicit expressions for the effective Hamiltonian up to fourth order in the hopping in terms of electron operators. In addition, more details are given for the second canonical transformation that is used to obtain the mapping of the $d = 0$ sector. The second order contribution to the low energy effective Hamiltonian reads

$$\sum_{\nu,\nu'} \hat{P}_0^\nu \hat{H}_{\text{eff}}^{(2)}(t)\hat{P}_0^{\nu'} = -\sum_{\substack{m,m'\\\nu,\nu'}} \frac{\hat{P}_0^\nu \hat{T}_m^{-1}\hat{P}_1^0 \hat{T}_{m'}^{+1}\hat{P}_0^{\nu'}}{U + J_H + m\omega} e^{i(m+m')\omega t}. \tag{61}$$

Note that for the $\hat{P}_1^{\nu}$ sector, only $\nu{=}0$ contributes since the $\hat{P}_1^1$ sector is not connected to $\hat{P}_{d\neq1}^{\nu}$ for inter-orbital hopping $t_{\alpha\neq\beta}{=}0$. The fourth order contribution to the low energy effective Hamiltonian reads:

$$\sum_{\nu\nu'}\hat{P}_0^{\nu}\big\{\hat{H}_{\mathrm{eff}}^{(4)}(t)+\tilde{H}_{\mathrm{eff}}^{(4)}(t)\big\}\hat{P}_0^{\nu'},\tag{62}$$

where the first term, $\hat{P}_0^{\nu}\hat{H}_{\mathrm{eff}}^{(4)}(t)\hat{P}_0^{\nu'}$, is computed using the direct generalized canonical transformation, Eq. (27) while the second term $\hat{P}_0^{\nu}\tilde{H}_{\mathrm{eff}}^{(4)}(t)\hat{P}_0^{\nu'}$ is computed using the second canonical transformation with terms $\hat{P}_0^{\nu}\hat{H}_{\mathrm{eff}}^{(2)}(t)\hat{P}_2^{\nu'}+h.c.$ in order to obtain the fourth order contribution to the effective Hamiltonian in the $d{=}0$ sector. After developing the commutator in Eq. (27) of the main text and inserting identities $\sum_{d,\nu}\hat{P}_d^{\nu}{=}1$, $\hat{P}_0^{\nu}\hat{H}_{\mathrm{eff}}^{(4)}(t)\hat{P}_0^{\nu'}$ reads

$$\sum_{\nu,\nu'}\hat{P}_0^{\nu}\hat{H}_{\mathrm{eff}}^{(4)}(t)\hat{P}_0^{\nu'}=\frac{1}{8}\sum_p\sum_{\substack{k+l+m+n=p}}\hat{P}_0^{\nu}\sum_{\substack{\nu,\nu',\nu''\\d=0,2}}\Big\{i\hat{S}_k^{(1)}\hat{P}_1^0 i\hat{S}_l^{(1)}\hat{P}_d^{\nu''}i\hat{S}_m^{(1)}\hat{P}_1^0\hat{T}_n^{+1}$$

$$-i\hat{S}_l^{(1)}\hat{P}_1^0 i\hat{S}_m^{(1)}\hat{P}_d^{\nu''}\hat{T}_n^{\pm1}\hat{P}_1^0 i\hat{S}_k^{(1)}-i\hat{S}_k^{(1)}\hat{P}_1^0 i\hat{S}_m^{(1)}\hat{P}_d^{\nu''}\hat{T}_n^{\pm1}\hat{P}_1^0 i\hat{S}_l^{(1)}-i\hat{S}_k^{(1)}\hat{P}_1^0 i\hat{S}_l^{(1)}\hat{P}_d^{\nu''}\hat{T}_n^{\pm1}\hat{P}_1^0 i\hat{S}_m^{(1)}$$

$$\tag{63}$$

$$+i\hat{S}_m^{(1)}\hat{P}_1^0\hat{T}_n^{\pm1}\hat{P}_d^{\nu''}i\hat{S}_l^{(1)}\hat{P}_1^0 i\hat{S}_k^{(1)}+i\hat{S}_l^{(1)}\hat{P}_1^0\hat{T}_n^{\pm1}\hat{P}_d^{\nu''}i\hat{S}_m^{(1)}\hat{P}_1^0 i\hat{S}_k^{(1)}+i\hat{S}_k^{(1)}\hat{P}_1^0\hat{T}_n^{\pm1}\hat{P}_d^{\nu''}i\hat{S}_m^{(1)}\hat{P}_1^0 i\hat{S}_l^{(1)}$$

$$-\hat{T}_n^{-1}\hat{P}_1^0 i\hat{S}_m^{(1)}\hat{P}_d^{\nu''}i\hat{S}_l^{(1)}\hat{P}_1^0 i\hat{S}_k^{(1)}\Big\}\hat{P}_0^{\nu'}e^{ip\omega t},$$

where $\hat{T}_m^{\pm1}=\hat{T}_m^{+1}+\hat{T}_m^{-1}$. We introduce the shorthand notations:

$$\hat{P}_0^{\nu}\hat{\mathbb{T}}_{kl}^{--}\hat{P}_2^{\nu'}=\hat{P}_0^{\nu}\hat{T}_k^{-1}\hat{P}_1^0\hat{T}_l^{-1}\hat{P}_2^{\nu'},\tag{64}$$

$$\hat{P}_2^{\nu}\hat{\mathbb{T}}_{kl}^{++}\hat{P}_0^{\nu'}=\hat{P}_2^{\nu}\hat{T}_k^{+1}\hat{P}_1^0\hat{T}_l^{+1}\hat{P}_0^{\nu'},\tag{65}$$

$$\hat{P}_0^{\nu}\hat{\mathbb{T}}_{kl}^{-+}\hat{P}_0^{\nu'}=\hat{P}_0^{\nu}\hat{T}_k^{-1}\hat{P}_1^0\hat{T}_l^{+1}\hat{P}_0^{\nu'}.\tag{66}$$

We express $i\hat{S}_m^{(1)}$ in terms of hopping operators $\hat{T}_m^{\pm1}$ and factors $C_{dd'}^{\nu\nu',m}$ using Eq. (22) in the main text. After simplification we obtain

$$\sum_{\nu,\nu'}\hat{P}_0^{\nu}\hat{H}_{\mathrm{eff}}^{(4)}(t)\hat{P}_0^{\nu'}=\sum_p\hat{P}_0^{\nu}\sum_{\substack{k+l+m+n=p\\\nu,\nu',\nu''}}\Big\{\frac{1}{4}C_{01}^{00,k}C_{12}^{00,l}(C_{21}^{00,m}-3C_{10}^{00,m})\hat{\mathbb{T}}_{kl}^{--}\hat{P}_2^0\hat{\mathbb{T}}_{mn}^{++}$$

$$+\frac{1}{8}\Big[C_{01}^{00,k}C_{12}^{01,l}\hat{\mathbb{T}}_{kl}^{--}\hat{P}_2^1(C_{21}^{10,m}\hat{\mathbb{T}}_{mn}^{++}-3C_{10}^{00,m}\hat{\mathbb{T}}_{nm}^{++})-C_{10}^{00,k}C_{21}^{10,l}(C_{12}^{01,m}\hat{\mathbb{T}}_{nm}^{--}-3C_{01}^{00,m}\hat{\mathbb{T}}_{mn}^{--})\hat{P}_2^1\hat{\mathbb{T}}_{lk}^{++}\Big]$$

$$+\frac{1}{2}C_{01}^{00,k}C_{10}^{00,l}C_{01}^{00,m}\big(\hat{\mathbb{T}}_{kl}^{-+}\hat{P}_0^{\nu''}\hat{\mathbb{T}}_{mn}^{-+}+\hat{\mathbb{T}}_{nm}^{-+}\hat{P}_0^{\nu''}\hat{\mathbb{T}}_{lk}^{-+}\big)\Big\}\hat{P}_0^{\nu'}e^{ip\omega t},\tag{67}$$

where we used $C_{d0}^{\nu0,m}{=}C_{d0}^{\nu1,m}$. Note that the last term of Eq. (67) describes the hopping process via $\hat{P}_0^{\nu}$ states and therefore, is simpler than the rest of the equation which describes hopping processes via $\hat{P}_2^0$ and $\hat{P}_2^1$. Performing the second canonical transformation we obtain:

$$\sum_{\nu,\nu'}\hat{P}_0^{\nu}\tilde{H}_{\mathrm{eff}}^{(4)}(t)\hat{P}_0^{\nu'}=\frac{1}{2}\sum_{\nu,\nu'}\sum_m\hat{P}_0^{\nu}\big[i\tilde{S}_m^{(1)}(t),\tilde{T}_{m'}^{\pm1}(t)\big]\hat{P}_0^{\nu'},\tag{68}$$

where $\hat{P}_d^\nu i\tilde{S}_m^{(1)}(t)\hat{P}_{d'}^{\nu'}=C_{dd'}^{\nu\nu',m}\hat{P}_d^\nu \tilde{T}_m^{\pm 1}(t)\hat{P}_{d'}^{\nu'}$, $C_{dd'}^{\nu\nu',m}=(E_d^\nu-E_{d'}^{\nu'}+m\omega)^{-1}$.

Note that $E_d^\nu=E_d^{\nu(0)}+E_d^{\nu(2)}$, where $E_d^{\nu(n)}$ is the energy contribution to $E_d^\nu$ of order $t_0^n$. We use $E_d^\nu\simeq E_d^{\nu(0)}$ and do not take into account second order contribution $E_d^{\nu(2)}$ since it leads to $6^{th}$ order corrections to $\hat{P}_0^\nu \tilde{H}_{\text{eff}}^{(4)}(t)\hat{P}_0^{\nu'}$.

The effective hoppings $\tilde{T}_m^{\pm 1}(t)$ in the second canonical transformation are determined by second order off-diagonal contributions to $\hat{H}_{\text{eff}}^{(2)}$:

$$\tilde{T}_m^{+1}(t) = \sum_{\nu\nu'}\hat{P}_2^\nu \hat{H}_{\text{eff},m}^{(2)}(t)\hat{P}_0^{\nu'}, \quad \tilde{T}_m^{-1}(t) = \sum_{\nu\nu'}\hat{P}_0^\nu \hat{H}_{\text{eff},m}^{(2)}(t)\hat{P}_2^{\nu'}. \tag{69}$$

Substitution of Eq. (25) yields

$$\tilde{T}_m^{+1}(t) = \frac{1}{2}\sum_{\nu\nu'}\sum_{k+l=m}\hat{P}_2^\nu \big(C_{01}^{00,k}\hat{\mathbb{T}}_{kl}^{++} - C_{12}^{0\nu,k}\hat{\mathbb{T}}_{lk}^{++}\big)\hat{P}_0^{\nu'} e^{im\omega t}, \tag{70}$$

and

$$\tilde{T}_m^{-1}(t) = -\frac{1}{2}\sum_{\nu\nu'}\sum_{k+l=m}\hat{P}_0^\nu \big(C_{21}^{\nu'0,k}\hat{\mathbb{T}}_{kl}^{--} - C_{10}^{00,k}\hat{\mathbb{T}}_{lk}^{--}\big)\hat{P}_2^{\nu'} e^{im\omega t}. \tag{71}$$

Note that $\hat{\mathbb{T}}_{kl}^{qq}\neq\hat{\mathbb{T}}_{lk}^{qq}$, for $k\neq l$. Finally, $\hat{P}_0^\nu \tilde{H}_{\text{eff}}^{(4)}(t)\hat{P}_0^{\nu'}$ reads

$$\sum_{\nu,\nu'}\hat{P}_0^\nu \tilde{H}_{\text{eff}}^{(4)}(t)\hat{P}_0^{\nu'} = \sum_p \frac{1}{8}\sum_{\substack{k+l+m+n=p\\ \nu,\nu',\nu''}}\hat{P}_0^\nu \big[C_{02}^{0\nu'',k+l}(C_{01}^{00,k}-C_{12}^{0\nu'',k})(C_{21}^{\nu''0,m}-C_{10}^{00,m})\hat{\mathbb{T}}_{kl}^{--}\hat{P}_2^{\nu''}\hat{\mathbb{T}}_{mn}^{++}$$
$$-C_{20}^{\nu''0,k+l}(C_{01}^{00,m}-C_{12}^{0\nu'',m})(C_{21}^{\nu''0,k}-C_{10}^{00,k})\hat{\mathbb{T}}_{mn}^{--}\hat{P}_2^{\nu''}\hat{\mathbb{T}}_{kl}^{++}\big]\hat{P}_0^{\nu'} e^{ip\omega t}. \tag{72}$$

## C  Spin-one and pseudo spin-one operators

Here we derive expressions for pseudo spin-one operators $\hat{\mathcal{T}}^q$, where $q=\pm 1, 0$. Starting from the spin-one operators in term of spin $1/2$ operators acting on orbital $\alpha,\beta=a,b$.

$$\hat{S}^{+1} = -\sum_{\alpha\neq\beta}(\hat{s}_\alpha^+\hat{s}_\alpha^- + \hat{s}_\alpha^-\hat{s}_\alpha^+)\hat{s}_\beta^+, \quad \hat{S}^{-1} = -\sum_{\alpha\neq\beta}(\hat{s}_\alpha^+\hat{s}_\alpha^- + \hat{s}_\alpha^-\hat{s}_\alpha^+)\hat{s}_\beta^-, \tag{73}$$

$$\hat{S}^0 = \hat{s}_a^+\hat{s}_a^-\hat{s}_b^+\hat{s}_b^- - \hat{s}_a^-\hat{s}_a^+\hat{s}_b^-\hat{s}_b^+, \tag{74}$$

where $\hat{s}_\alpha^+=\hat{c}_{\alpha\uparrow}^\dagger \hat{c}_{\alpha\downarrow}$ and $\hat{s}_\alpha^-=\hat{c}_{\alpha\downarrow}^\dagger \hat{c}_{\alpha\uparrow}$, we obtain anolog expressions for pseudo-spin one operators $\hat{\mathcal{T}}^q$ in terms of the Anderson pseudo-spin $1/2$ operators $\hat{\tau}^\pm$ [43]. Using $\hat{\tau}_\alpha^+=\hat{c}_{\alpha\uparrow}^\dagger \hat{c}_{\alpha\downarrow}^\dagger$ and $\hat{\tau}_\alpha^-=\hat{c}_{\alpha\downarrow}\hat{c}_{\alpha\uparrow}$, we have

$$\hat{\mathcal{T}}^{+1} = -\sum_{\alpha\neq\beta}(\hat{\tau}_\alpha^+\hat{\tau}_\alpha^- + \hat{\tau}_\alpha^-\hat{\tau}_\alpha^+)\hat{\tau}_\beta^+, \quad \hat{\mathcal{T}}^{-1} = -\sum_{\alpha\neq\beta}(\hat{\tau}_\alpha^+\hat{\tau}_\alpha^- + \hat{\tau}_\alpha^-\hat{\tau}_\alpha^+)\hat{\tau}_\beta^-, \tag{75}$$

$$\hat{\mathcal{T}}^0 = \hat{\tau}_a^+\hat{\tau}_a^-\hat{\tau}_b^+\hat{\tau}_b^- - \hat{\tau}_a^-\hat{\tau}_a^+\hat{\tau}_b^-\hat{\tau}_b^+. \tag{76}$$

Hence, in terms of electron operators the pseudo-spin one operators read

$$\hat{\mathcal{T}}^{+1} = -\sum_{\alpha\neq\beta}(\hat{n}_{\alpha\uparrow}\hat{n}_{\alpha\downarrow} + \hat{h}_{\alpha\uparrow}\hat{h}_{\alpha\downarrow})\hat{c}_{\beta\uparrow}^\dagger \hat{c}_{\beta\downarrow}^\dagger, \quad \hat{\mathcal{T}}^{-1} = -\sum_{\alpha\neq\beta}(\hat{n}_{\alpha\uparrow}\hat{n}_{\alpha\downarrow} + \hat{h}_{\alpha\uparrow}\hat{h}_{\alpha\downarrow})\hat{c}_{\beta\downarrow}\hat{c}_{\beta\uparrow}, \tag{77}$$

$$\hat{\mathcal{T}}^0 = \hat{n}_{a\uparrow}\hat{n}_{a\downarrow}\hat{n}_{b\uparrow}\hat{n}_{b\downarrow} + \hat{h}_{a\uparrow}\hat{h}_{a\downarrow}\hat{h}_{b\uparrow}\hat{h}_{b\downarrow}. \tag{78}$$

# D   Derivation of the effective spin-one Hamiltonian

Here we derive the effective Hamiltonian up to fourth order in the hopping in terms of spin-one operators. First we describe the second-order effective Hamiltonian $\sum_{\nu,\nu'} \hat{P}_0^\nu \hat{H}_{\text{eff}}^{(2)} \hat{P}_0^{\nu'}$ in terms of spin-one operators and $\hat{R}_{ij}^1$. Second, we describe the contribution to the fourth-order effective Hamiltonian $\sum_{\nu,\nu'} \hat{P}_0^\nu \{\hat{H}_{\text{eff}}^{(4)} + \tilde{H}_{\text{eff}}^{(4)}\} \hat{P}_0^{\nu'}$ obtained with the first and the second canonical transformation in terms of spin-one operators, $\hat{R}_{ij}^1$ and $\hat{R}_{ij}^2$. Third, we do the third canonical transformation which takes a state from $P_0^\nu$ sector as a high energy state. Since the low-energy subspace contains not only spin-one terms, we perform an additional downfolding.

The bilinear spin-one term reads

$$
\vec{S}_i \cdot \vec{S}_j = \frac{1}{2} \sum_{\alpha \neq \beta, \sigma} \Big\{ \hat{c}_{i\alpha\sigma}^\dagger \hat{c}_{i\alpha\bar{\sigma}} \hat{c}_{j\alpha\bar{\sigma}}^\dagger \hat{c}_{j\alpha\sigma} \big( \hat{h}_{i\beta\sigma} \hat{n}_{i\beta\bar{\sigma}} \hat{n}_{j\beta\sigma} \hat{h}_{j\beta\bar{\sigma}} + \hat{n}_{i\beta\sigma} \hat{h}_{i\beta\bar{\sigma}} \hat{h}_{j\beta\sigma} \hat{n}_{j\beta\bar{\sigma}} \big) + \hat{c}_{i\alpha\sigma}^\dagger \hat{c}_{i\alpha\bar{\sigma}} \hat{c}_{j\beta\bar{\sigma}}^\dagger \hat{c}_{j\beta\sigma}
$$
$$
\times \big( \hat{n}_{i\beta\sigma} \hat{h}_{i\beta\bar{\sigma}} \hat{h}_{j\alpha\sigma} \hat{n}_{j\alpha\bar{\sigma}} + \hat{h}_{i\beta\sigma} \hat{n}_{i\beta\bar{\sigma}} \hat{n}_{j\alpha\sigma} \hat{h}_{j\alpha\bar{\sigma}} \big) - \hat{n}_{i\alpha\sigma} \hat{h}_{i\alpha\bar{\sigma}} \hat{n}_{i\beta\sigma} \hat{h}_{i\beta\bar{\sigma}} \hat{h}_{j\alpha\sigma} \hat{n}_{j\alpha\bar{\sigma}} \hat{h}_{j\beta\sigma} \hat{n}_{j\beta\bar{\sigma}} \Big\}. \tag{79}
$$

The first two terms of $\vec{S}_i \cdot \vec{S}_j$ allows an exchange interaction process that transforms a state that does not violate Hund rule ($\nu=0$) to a state which violates Hund rule ($\nu=1$) and vice versa. The last term is a density term which stands for $\hat{S}_i^0 \hat{S}_j^0$. One can derive the following relation

$$
\sum_{\substack{m \\ \nu,\nu'}} \hat{P}_0^\nu \hat{T}_m^- \hat{P}_1^0 \hat{T}_{-m}^+ \hat{P}_0^{\nu'} = t_0^2 \sum_m J_{|m|}^2(\mathcal{E}) \sum_{<i,j>} \big( \vec{S}_i \cdot \vec{S}_j + \hat{R}_{ij}^1 \big), \tag{80}
$$

and use it to write $\hat{P}_0^\nu \hat{H}_{\text{eff}}^{(2)}(t) \hat{P}_0^{\nu'}$ in terms of $\vec{S}_i \cdot \vec{S}_j$. After time averaging $\bar{H}_{\text{eff}}^{(2)} = \frac{1}{T} \int_0^T \hat{H}_{\text{eff}}^{(2)}(t) dt$, we obtain:

$$
\sum_{\nu,\nu'} \hat{P}_0^\nu \bar{H}_{\text{eff}}^{(2)} \hat{P}_0^{\nu'} = \sum_{<i,j>} J_{ex} \big\{ \vec{S}_i \cdot \vec{S}_j + \hat{R}_{ij}^1 \big\}. \tag{81}
$$

The term $\hat{R}_{ij}^1$ contains a description for the non spin-one states from $\hat{P}_0^\nu$, the coupling between a $S=1$ and a $S \neq 1$ state and a constant term. All these features can be easily seen after the rotation of $\hat{R}_{ij}^1$ into the spin-one basis. Here we restrict to $\hat{R}_{ij}^1$ written in terms of single electron operators:

$$
\hat{R}_{ij}^1 = \frac{1}{2} \sum_{\alpha \neq \beta, \sigma} \Big\{ \hat{c}_{i\alpha\sigma}^\dagger \hat{c}_{i\alpha\bar{\sigma}} \hat{c}_{j\alpha\bar{\sigma}}^\dagger \hat{c}_{j\alpha\sigma} \big( \hat{h}_{i\beta\sigma} \hat{n}_{i\beta\bar{\sigma}} \hat{n}_{j\beta\sigma} \hat{h}_{j\beta\bar{\sigma}} + \hat{n}_{i\beta\sigma} \hat{h}_{i\beta\bar{\sigma}} \hat{h}_{j\beta\sigma} \hat{n}_{j\beta\bar{\sigma}} \big)
$$
$$
- \hat{c}_{i\alpha\sigma}^\dagger \hat{c}_{i\alpha\bar{\sigma}} \hat{c}_{j\beta\bar{\sigma}}^\dagger \hat{c}_{j\beta\sigma} \big( \hat{n}_{i\beta\sigma} \hat{h}_{i\beta\bar{\sigma}} \hat{h}_{j\alpha\sigma} \hat{n}_{j\alpha\bar{\sigma}} + \hat{h}_{i\beta\sigma} \hat{n}_{i\beta\bar{\sigma}} \hat{n}_{j\alpha\sigma} \hat{h}_{j\alpha\bar{\sigma}} \big) \Big\} \tag{82}
$$
$$
- \sum_\sigma \big\{ \hat{n}_{i a\sigma} \hat{h}_{i a\bar{\sigma}} \hat{n}_{i b\sigma} \hat{h}_{i b\bar{\sigma}} \hat{h}_{j a\sigma} \hat{n}_{j a\bar{\sigma}} \hat{n}_{j b\sigma} \hat{n}_{j b\bar{\sigma}} + 2 \hat{n}_{i a\sigma} \hat{h}_{i a\bar{\sigma}} \hat{h}_{i b\sigma} \hat{n}_{i b\bar{\sigma}} \hat{h}_{j a\sigma} \hat{n}_{j a\bar{\sigma}} \hat{n}_{j b\sigma} \hat{h}_{j b\bar{\sigma}} \big\}.
$$

Similarly, we map $\hat{P}_0\tilde{H}^{(4)}_{\text{eff}}\hat{P}_0$ onto the spin-one model. The biquadratic spin-one term reads

$$
\begin{aligned}
(\vec{S}_i\cdot\vec{S}_j)^2 = \sum_{\alpha\neq\beta,\sigma}\Big\{ &-\frac{1}{2}\big[\hat{c}^\dagger_{ia\sigma}\hat{c}_{ia\bar{\sigma}}\hat{c}^\dagger_{ja\bar{\sigma}}\hat{c}_{ja\sigma}\big(\hat{h}_{i\beta\sigma}\hat{n}_{i\beta\bar{\sigma}}\hat{n}_{j\beta\sigma}\hat{h}_{j\beta\bar{\sigma}}+\hat{n}_{i\beta\sigma}\hat{h}_{i\beta\bar{\sigma}}\hat{h}_{j\beta\sigma}\hat{n}_{j\beta\bar{\sigma}}\big) \\
&+\hat{c}^\dagger_{ia\sigma}\hat{c}_{ia\bar{\sigma}}\hat{c}^\dagger_{j\beta\bar{\sigma}}\hat{c}_{j\beta\sigma}\big(\hat{n}_{i\beta\sigma}\hat{h}_{i\beta\bar{\sigma}}\hat{h}_{ja\sigma}\hat{n}_{ja\bar{\sigma}}+\hat{h}_{i\beta\sigma}\hat{n}_{i\beta\bar{\sigma}}\hat{n}_{ja\sigma}\hat{h}_{ja\bar{\sigma}}\big)\big] \\
&+\hat{n}_{ia\sigma}\hat{h}_{ia\bar{\sigma}}\hat{n}_{i\beta\sigma}\hat{h}_{i\beta\bar{\sigma}}\hat{h}_{ja\sigma}\hat{n}_{ja\bar{\sigma}}\hat{h}_{j\beta\sigma}\hat{n}_{j\beta\bar{\sigma}}\Big\} \\
&+\frac{1}{2}\sum_\sigma\Big\{2\hat{c}^\dagger_{ia\sigma}\hat{c}_{ia\bar{\sigma}}\hat{c}^\dagger_{ja\bar{\sigma}}\hat{c}_{ja\sigma}\hat{c}^\dagger_{ib\sigma}\hat{c}_{ib\bar{\sigma}}\hat{c}^\dagger_{jb\bar{\sigma}}\hat{c}_{jb\sigma} \\
&+\hat{c}^\dagger_{ia\sigma}\hat{c}_{ia\bar{\sigma}}\hat{c}^\dagger_{ib\bar{\sigma}}\hat{c}_{ib\sigma}\big(\hat{c}^\dagger_{ja\bar{\sigma}}\hat{c}_{ja\sigma}\hat{c}^\dagger_{jb\sigma}\hat{c}_{jb\bar{\sigma}}+\hat{c}^\dagger_{ja\sigma}\hat{c}_{ja\bar{\sigma}}\hat{c}^\dagger_{jb\bar{\sigma}}\hat{c}_{jb\sigma}\big) \\
&+\hat{n}_{ia\sigma}\hat{h}_{ia\bar{\sigma}}\hat{h}_{ib\sigma}\hat{n}_{ib\bar{\sigma}}\big(\hat{h}_{ja\sigma}\hat{n}_{ja\bar{\sigma}}\hat{n}_{jb\sigma}\hat{h}_{jb\bar{\sigma}}+\hat{n}_{ja\sigma}\hat{h}_{ja\bar{\sigma}}\hat{h}_{jb\sigma}\hat{n}_{jb\bar{\sigma}}\big) \\
&+\big[\hat{c}^\dagger_{ia\sigma}\hat{c}_{ia\bar{\sigma}}\hat{c}^\dagger_{ib\bar{\sigma}}\hat{c}_{ib\sigma}\big(\hat{h}_{ja\sigma}\hat{n}_{ja\bar{\sigma}}\hat{h}_{jb\sigma}\hat{n}_{jb\bar{\sigma}}+\hat{n}_{ja\sigma}\hat{h}_{ja\bar{\sigma}}\hat{n}_{jb\sigma}\hat{h}_{jb\bar{\sigma}}\big)+h.c.\big]\Big\}.
\end{aligned}
\tag{83}
$$

The first summation in Eq. (83) contains terms which are very similar to $\vec{S}_i\cdot\vec{S}_j$ *i.e.* it contains density terms which describe the states in the $\hat{P}^0_0$ sector and terms which connect the $\hat{P}^0_0$ states to the $\hat{P}^1_0$ states. The second summation contains density terms which describe the states in the $\hat{P}^1_0$ and terms wich operate an internal mixing of the $\hat{P}^0_0$ states as well as an internal mixing of the $\hat{P}^1_0$ states. From Eq. (67) we obtain the following equalities

$$
\begin{aligned}
\sum_{\substack{k,l,m,n \\ \nu,\nu',\nu''}}\hat{P}^\nu_0\hat{\mathbb{T}}^{-+}_{kl}\hat{P}^{\nu''}_0\hat{\mathbb{T}}^{-+}_{mn}\hat{P}^{\nu'}_0=&t^4_0\sum_{k,l,m,n}(-1)^k J_k(\mathcal{E})J_l(\mathcal{E})J_m(\mathcal{E})J_n(\mathcal{E}) \\
&\times\big[(-1)^m+(-1)^n\big]\sum_{<i,j>}\big(\vec{S}_i\cdot\vec{S}_j+\hat{R}^1_{ij}\big)^2,
\end{aligned}
\tag{84}
$$

for the hopping process via $\hat{P}^{\nu''}_0$ sector,

$$
\sum_{\substack{k,l,m,n \\ \nu,\nu'}}\hat{P}^\nu_0\hat{\mathbb{T}}^{--}_{kl}\hat{P}^0_2\hat{\mathbb{T}}^{++}_{mn}\hat{P}^{\nu'}_0=4t^4_0\sum_{k,l,m,n}(-1)^{k+l}J_k(\mathcal{E})J_l(\mathcal{E})J_m(\mathcal{E})J_n(\mathcal{E})\sum_{<i,j>}\big((\vec{S}_i\cdot\vec{S}_j)^2+\hat{R}^2_{ij}\big),
\tag{85}
$$

for the hopping process via $\hat{P}^0_2$ sector,

$$
\begin{aligned}
\sum_{\substack{k,l,m,n \\ \nu,\nu'}}\hat{P}^\nu_0\hat{\mathbb{T}}^{--}_{kl}\hat{P}^1_2\hat{\mathbb{T}}^{++}_{mn}\hat{P}^{\nu'}_0=&2t^4_0\sum_{k,l,m,n}(-1)^k J_k(\mathcal{E})J_l(\mathcal{E})J_m(\mathcal{E})J_n(\mathcal{E}) \\
&\times\big[(-1)^m+(-1)^n\big]\sum_{<i,j>}\big((\vec{S}_i\cdot\vec{S}_j)^2+\hat{R}^2_{ij}\big),
\end{aligned}
\tag{86}
$$

for the hopping process via $\hat{P}_2^1$ sector. We obtain the following expression for $\hat{R}_{ij}^2$

$$
\begin{aligned}
\hat{R}_{ij}^2 = \frac{1}{2}\sum_{\alpha\neq\beta,\sigma}\Big\{ &-\hat{c}_{i\alpha\sigma}^\dagger\hat{c}_{i\alpha\bar{\sigma}}\hat{c}_{j\alpha\bar{\sigma}}^\dagger\hat{c}_{j\alpha\sigma}\big(\hat{h}_{i\beta\sigma}\hat{n}_{i\beta\bar{\sigma}}\hat{n}_{j\beta\sigma}\hat{h}_{j\beta\bar{\sigma}}+\hat{n}_{i\beta\sigma}\hat{h}_{i\beta\bar{\sigma}}\hat{h}_{j\beta\sigma}\hat{n}_{j\beta\bar{\sigma}}\big) \\
&+\hat{c}_{i\alpha\sigma}^\dagger\hat{c}_{i\alpha\bar{\sigma}}\hat{c}_{j\beta\bar{\sigma}}^\dagger\hat{c}_{j\beta\sigma}\big(\hat{n}_{i\beta\sigma}\hat{h}_{i\beta\bar{\sigma}}\hat{h}_{j\alpha\sigma}\hat{n}_{j\alpha\bar{\sigma}}+\hat{h}_{i\beta\sigma}\hat{n}_{i\beta\bar{\sigma}}\hat{n}_{j\alpha\sigma}\hat{h}_{j\alpha\bar{\sigma}}\big) \\
&-\hat{n}_{i\alpha\sigma}\hat{h}_{i\alpha\bar{\sigma}}\hat{n}_{i\beta\sigma}\hat{h}_{i\beta\bar{\sigma}}\hat{h}_{j\alpha\sigma}\hat{n}_{j\alpha\bar{\sigma}}\hat{h}_{j\beta\sigma}\hat{n}_{j\beta\bar{\sigma}}\Big\} \\
+\frac{1}{2}\sum_{\sigma}\Big\{ &\hat{c}_{i\alpha\sigma}^\dagger\hat{c}_{i\alpha\bar{\sigma}}\hat{c}_{ib\bar{\sigma}}^\dagger\hat{c}_{ib\sigma}\big(\hat{c}_{j\alpha\bar{\sigma}}^\dagger\hat{c}_{j\alpha\sigma}\hat{c}_{jb\bar{\sigma}}^\dagger\hat{c}_{jb\sigma}-\hat{c}_{j\alpha\sigma}^\dagger\hat{c}_{j\alpha\bar{\sigma}}\hat{c}_{jb\sigma}^\dagger\hat{c}_{jb\bar{\sigma}}\big) \\
&+\hat{n}_{i\alpha\sigma}\hat{h}_{i\alpha\bar{\sigma}}\hat{h}_{ib\sigma}\hat{n}_{ib\bar{\sigma}}\big(\hat{h}_{j\alpha\sigma}\hat{n}_{j\alpha\bar{\sigma}}\hat{n}_{jb\sigma}\hat{h}_{jb\bar{\sigma}}-\hat{n}_{j\alpha\sigma}\hat{h}_{j\alpha\bar{\sigma}}\hat{h}_{jb\sigma}\hat{n}_{jb\bar{\sigma}}\big) \\
&-\big[\hat{c}_{i\alpha\sigma}^\dagger\hat{c}_{i\alpha\bar{\sigma}}\hat{c}_{ib\sigma}^\dagger\hat{c}_{ib\bar{\sigma}}\big(\hat{h}_{j\alpha\sigma}\hat{n}_{j\alpha\bar{\sigma}}\hat{h}_{jb\sigma}\hat{n}_{jb\bar{\sigma}}+\hat{n}_{j\alpha\sigma}\hat{h}_{j\alpha\bar{\sigma}}\hat{n}_{jb\sigma}\hat{h}_{jb\bar{\sigma}}\big)+\text{h.c.}\big]\Big\}.
\end{aligned}
\tag{87}
$$

$\hat{R}_{ij}^2$ rotated into the spin-one basis contains a fourth order contribution to the energy of the non spin-one states, a contribution to the coupling between the spin-one and the non spin-one state and the same constant as in $\hat{R}_{ij}^1$.

We use these equalities to write the time averaged effective Hamiltonian in terms of spin-one operators

$$
\begin{aligned}
\bar{H}_{\text{eff}}^{d=0} = \sum_{<i,j>}\Big\{ &K_1(\mathcal{E},\omega)\big(\vec{S}_i\cdot\vec{S}_j+\hat{R}_{ij}^1\big)+K_2(\mathcal{E},\omega)\big(\vec{S}_i\cdot\vec{S}_j+\hat{R}_{ij}^1\big)^2 \\
&+K_3(\mathcal{E},\omega)\big((\vec{S}_i\cdot\vec{S}_j)^2+\hat{R}_{ij}^2\big)\Big\},
\end{aligned}
\tag{88}
$$

where $K_1(\mathcal{E},\omega)=J_{\text{ex}}(\mathcal{E},\omega)$ is the second order Heisenberg exchange interaction and reads

$$
K_1(\mathcal{E},\omega)=\sum_{m=-\infty}^{+\infty}\frac{t_0^2 J_{|m|}^2(\mathcal{E})}{U+J_H+m\omega}.
\tag{89}
$$

$K_2(\mathcal{E},\omega)$ is responsible for an energy contribution to both the fourth order Heisenberg exchange as well as the biquadratic exchange interaction, see Eqs. (38) and (40) of the main text.

$$
K_2(\mathcal{E},\omega)=t_0^4\sum_{k+l+m+n=0}(-1)^k J_k(\mathcal{E})J_l(\mathcal{E})J_m(\mathcal{E})J_n(\mathcal{E})\big[(-1)^m+(-1)^n\big]C_{01}^{00,k}C_{10}^{00,l}C_{01}^{00,m}.
\tag{90}
$$

$K_3(\mathcal{E},\omega)$ only enters in the biquadratic exchange interaction formula, see Eq. (40), and reads

$$
K_3(\mathcal{E},\omega)=t_0^4\sum_{k+l+m+n=0}(-1)^k J_k(\mathcal{E})J_l(\mathcal{E})J_m(\mathcal{E})J_n(\mathcal{E})\Big\{(-1)^l\mathbb{C}_2^0+\big[(-1)^m+(-1)^n\big]\frac{\mathbb{C}_2^1}{2}\Big\},
\tag{91}
$$

where the $\mathbb{C}_2^\gamma$ coefficients read

$$
\mathbb{C}_2^\gamma=C_{01}^{00,k}C_{12}^{0\gamma,l}(C_{21}^{\gamma 0,m}-3C_{10}^{00,m})+C_{02}^{0\gamma,k+l}(C_{01}^{00,k}-C_{12}^{0\gamma,k})(C_{21}^{\gamma 0,m}-C_{10}^{00,m}),
\tag{92}
$$

We now interest ourselves to the coupling between the spin-one and the non spin-one state of $\hat{P}_0^\gamma$ that is described by $\hat{R}_{ij}^1$. The basis transformation which allows one to go from a electron occupation number basis to the angular momentum basis is the following

$$|S, M_S, S_i, S_j\rangle = \sum_{M_i, M_j} C_{S_i M_i, S_j M_j}^{SM} |S_i, M_i, S_j, M_j\rangle, \tag{93}$$

where $C_{S_i M_i, S_j M_j}^{SM_S}$ are Clebsch-Gordan coefficients. From this basis transformation, we obtain three spin-one states, namely a singlet ($S=0$), a triplet ($S=1$) and a quintet state ($S=2$)

$$
\begin{aligned}
|0, 0, 1, 1\rangle = \frac{1}{2\sqrt{3}}\Big\{ & 2|\uparrow,\uparrow\rangle_i|\downarrow,\downarrow\rangle_j + 2|\downarrow,\downarrow\rangle_i|\uparrow,\uparrow\rangle_j - |\uparrow,\downarrow\rangle_i|\uparrow,\downarrow\rangle_j - |\downarrow,\uparrow\rangle_i|\downarrow,\uparrow\rangle_j \\
& -|\uparrow,\downarrow\rangle_i|\downarrow,\uparrow\rangle_j - |\downarrow,\uparrow\rangle_i|\uparrow,\downarrow\rangle_j\Big\},
\end{aligned}
\tag{94}
$$

$$|1, 0, 1, 1\rangle = \frac{1}{\sqrt{2}}\Big\{ -|\uparrow,\uparrow\rangle_i|\downarrow,\downarrow\rangle_j + |\downarrow,\downarrow\rangle_i|\uparrow,\uparrow\rangle_j\Big\}, \tag{95}$$

and

$$
\begin{aligned}
|2, 0, 1, 1\rangle = \frac{1}{\sqrt{6}}\Big\{ & |\uparrow,\uparrow\rangle_i|\downarrow,\downarrow\rangle_j + |\downarrow,\downarrow\rangle_i|\uparrow,\uparrow\rangle_j + |\uparrow,\downarrow\rangle_i|\uparrow,\downarrow\rangle_j + |\downarrow,\uparrow\rangle_i|\downarrow,\uparrow\rangle_j \\
& + |\uparrow,\downarrow\rangle_i|\downarrow,\uparrow\rangle_j + |\downarrow,\uparrow\rangle_i|\uparrow,\downarrow\rangle_j\Big\},
\end{aligned}
\tag{96}
$$

and three states which are non spin-one states. We define spin-one projection operators such that $\sum_\nu \hat{P}_0^\gamma = \hat{\mathbb{P}}_S + \hat{\mathbb{P}}_R$, where subscripts $S$ and $R$ refere to $S=1$ and $S\neq1$ states respectively [34]. The third canonical transformation leads to the following fourth order contribution to the effective Hamiltonian

$$\tilde{\tilde{H}}_{\text{eff}}^{(4)}(t) = -\sum_{m,m'} \frac{\hat{\mathbb{P}}_S \hat{H}_{\text{eff},m}^{(2)} \hat{\mathbb{P}}_R \hat{H}_{\text{eff},m'}^{(2)} \hat{\mathbb{P}}_S}{E_R - E_S + m\omega} e^{i(m+m')\omega t}, \tag{97}$$

where, $E_S$ and $E_R$ are energies of the $S=1$ and $S\neq1$ states, respectively. The coupling between the $\hat{\mathbb{P}}_S$ and $\hat{\mathbb{P}}_R$ subspaces only involves the singlet state $|0, 0, 1, 1\rangle$ and the $S\neq1$ state $|0, 0, 0, 0\rangle$

$$|0, 0, 0, 0\rangle = \frac{1}{2}\Big\{ |\uparrow,\downarrow\rangle_i|\uparrow,\downarrow\rangle_j + |\downarrow,\uparrow\rangle_i|\downarrow,\uparrow\rangle_j - |\uparrow,\downarrow\rangle_i|\downarrow,\uparrow\rangle_j - |\downarrow,\uparrow\rangle_i|\uparrow,\downarrow\rangle_j\Big\}. \tag{98}$$

Note that, the energy approximation of Eq. (24) is only for $d\neq d'$. Here, within the $d=0$ sector, we take the exact value for the energies of $|0, 0, 1, 1\rangle$ and $|0, 0, 0, 0\rangle$, 0 and $4J_{\text{H}}$ respectively. After time averaging and projection onto the singlet state, yields

$$\tilde{\tilde{E}}_{\text{Singlet}}^{(4)} = -\frac{3}{2} t_0^4 \sum_{k+l+m+n=0} J_k(\mathcal{E}) J_l(\mathcal{E}) J_m(\mathcal{E}) J_n(\mathcal{E}) (C_{01}^{00,k} - C_{10}^{00,l})(C_{01}^{00,m} - C_{10}^{00,n}) \frac{(-1)^k((-1)^m + (-1)^n)}{(4J_H + (k+l)\omega)}. \tag{99}$$

This is the additional fourth order energy contribution to the singlet state due to the additional downfolding of $\hat{P}_0^\gamma$. Using the spin-one Hamiltonian

$$\hat{H}_{\text{spin}} = E_0 + \sum_{<i,j>} \Big\{ J_{\text{ex}}\vec{S}_i \cdot \vec{S}_j + B_{\text{ex}}(\vec{S}_i \cdot \vec{S}_j)^2 \Big\}, \tag{100}$$

one can obtain a relation between $J_{\text{ex}}$ and $B_{\text{ex}}$ and the spin-one state energies

$$J_{\text{ex}}=(E_{\text{Quintet}}-E_{\text{Triplet}})/4, \tag{101}$$

$$B_{\text{ex}}=(E_{\text{Quintet}}-E_{\text{Triplet}})/4-(E_{\text{Quintet}}-E_{\text{Singlet}})/6. \tag{102}$$

Since the additional downfolding of $\hat{P}_0^\nu$ gives rise to an energy contribution for the singlet state only, we have the additional energy contribution to the biquadratic exchange interaction $B_{\text{ex}}^{[3]}(\hat{P}_0)=\tilde{\tilde{E}}_{\text{Singlet}}^{(4)}/6$, which leads to

$$B_{\text{ex}}^{[3]}(\hat{P}_0) = -\sum_{k+l+m+n=0} A_{klmn}^1(\mathcal{E}) \frac{(C_{01}^{00,k}-C_{10}^{00,l})(C_{01}^{00,m}-C_{10}^{00,n})}{4(4J_H + (k+l)\omega)}, \tag{103}$$

where

$$A_{klmn}^1(\mathcal{E}) = t_0^4(-1)^k\big[(-1)^m+(-1)^n\big]J_k(\mathcal{E})J_l(\mathcal{E})J_m(\mathcal{E})J_n(\mathcal{E}). \tag{104}$$

Note that $B_{\text{ex}}^{[3]}(\hat{P}_0)\propto J_{\text{ex}}(\mathcal{E},\omega)/J_H$, which means that the canonical transformation gives an accurate description of $B_{\text{ex}}^{[3]}(\hat{P}_0)$ as long as $J_{\text{ex}}(\mathcal{E},\omega)\ll J_H$. This inequality is always fulfilled for the regime of frequencies studied here but breaks down when orbital resolved spin dynamics is needed, see the discussion in Section 3.1.

# E   Effective Hamiltonian with Spin-Charge coupling

In this section we discuss in more detail the effective Hamiltonian which describes the spin-charge coupling phenomenon. We study the effective Hamiltonian responsible for the non-equilibrium spin-charge coupling between $\hat{P}_0^\nu$ states from Fourier sector $m=0$ and $\hat{P}_2^0$ states from the $m=-1$ sector as shown in Figure 5. This effective Hamiltonian forms a 8×8 matrix in the occupation number basis states $|\phi_k\rangle$ of the $\hat{P}_0^\nu+\hat{P}_2^0$ sector, which yields the following matrix structure

$$\begin{pmatrix}
-4t_0^2F & 0 & 0 & 0 & 2t_0^2F & 2t_0^2F & \vdots & -t_0^2I & t_0^2I \\
0 & -4t_0^2F & 0 & 0 & 2t_0^2F & 2t_0^2F & \vdots & -t_0^2I & t_0^2I \\
0 & 0 & 2J_H & 0 & -J_H & -J_H & \vdots & 0 & 0 \\
0 & 0 & 0 & 2J_H & -J_H & -J_H & \vdots & 0 & 0 \\
2t_0^2F & 2t_0^2F & -J_H & -J_H & 2J_H-4t_0^2F & 0 & \vdots & t_0^2I & -t_0^2I \\
2t_0^2F & 2t_0^2F & -J_H & -J_H & 0 & 2J_H-4t_0^2F & \vdots & t_0^2I & -t_0^2I \\
\cdots & \cdots & \cdots & \cdots & \cdots & \cdots & \cdots & \cdots & \cdots \\
-t_0^2I^* & -t_0^2I^* & 0 & 0 & t_0^2I^* & t_0^2I^* & \vdots & E_I-\omega+4t_0^2G & 0 \\
t_0^2I^* & t_0^2I^* & 0 & 0 & -t_0^2I^* & -t_0^2I^* & \vdots & 0 & E_I-\omega+4t_0^2G
\end{pmatrix}, \tag{105}$$

where

$$F = \sum_m \frac{J_{|m|}(\mathcal{E})^2}{2(U + J_H + m\omega)}, \tag{106}$$

$$G = \sum_m \frac{J_{|m|}(\mathcal{E})^2}{2(3U - 5J_H + m\omega)}, \tag{107}$$

$$I = I(t) = \sum_k (-1)^k J_k(\mathcal{E}) J_{k+1}(\mathcal{E}) \left\{ \frac{1}{3U - 5J_H + k\omega} - \frac{1}{U + J_H + k\omega} \right\} e^{-i\omega t}. \tag{108}$$

$I^*$ is the complex conjugate of $I$ and $E_I = 4(U - J_H)$ is the energy of the doubly ionized states. The upper left block of the matrix Eq. (105) corresponds to the effective Hamiltonian in $\hat{P}_0^v$ sector with $m = 0$ where the basis states are

$$h_{d=0} = \Big\{ |\uparrow, \uparrow \rangle_i |\downarrow, \downarrow \rangle_j, |\downarrow, \downarrow \rangle_i |\uparrow, \uparrow \rangle_j, |\uparrow, \downarrow \rangle_i |\uparrow, \downarrow \rangle_j,$$
$$|\downarrow, \uparrow \rangle_i |\downarrow, \uparrow \rangle_j, |\uparrow, \downarrow \rangle_i |\downarrow, \uparrow \rangle_j, |\downarrow, \uparrow \rangle_i |\uparrow, \downarrow \rangle_j \Big\}, \tag{109}$$

where $|\sigma_a, \sigma'_b \rangle_i = \hat{c}^\dagger_{ib\sigma'} \hat{c}^\dagger_{ia\sigma} |0\rangle$. The lower right block of the matrix corresponds to the effective Hamiltonian in $\hat{P}_2^0$ sector with $m = -1$ where the basis is the following

$$h_{d=2} = \Big\{ |\uparrow\downarrow, \uparrow\downarrow \rangle_i |0,0\rangle_j, |0,0\rangle_i |\uparrow\downarrow, \uparrow\downarrow \rangle_j \Big\}. \tag{110}$$

We diagonalize the matrix, Eq. (105) and, after time averaging, obtain the spectrum shown in Figure 5. We emphasize that the diagonalization and time averaging do not commute e.g. $\frac{1}{T} \int_0^T I(t) dt = 0$ which washes out the coupling.

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
