# Peer review of "Optical control of competing exchange interactions and coherent spin-charge coupling in two-orbital Mott insulators"

_SciPost Physics, doi:SciPost Phys. 6, 027 (2019)_

## Round 1 · Referee Report · Marin Bukov · 2018-4-28

Strengths

1- the paper provides a careful and very detailed derivation of the effective Hamiltonian for the strongly-interacting two-orbital Fermi-Hubbard model under a periodic drive.

2- the paper contains useful remarks throughout to explain the details of the derivation, including appendices.

Weaknesses

1- the paper is partially technical and (especially in the first part after the introduction) might not be easy to grasp by inexperienced readers, although this seems justified by the subject of study.

Report

The manuscript "Optical control of competing exchange interactions and coherent spin-charge coupling in two-orbital Mott insulators" by Barbeau et al, studies the periodically-driven two-orbital Fermi Hubbard model. The paper derives the leasing-order effective Hamiltonian for the problem in the limit of strong interactions. The authors find that the presence of orbital dynamics and degrees of freedom does not change the control of the effective exchange interaction to leading order in the inverse interaction strength. At the same time, orbital dynamics gives access to charge states that are not accessible in equilibrium. This is demonstrated numerically in a two-site system using a Gaussian pulse signal.

I recommend publication in SciPost, once the following suggestions/questions have been considered.

1- In Eq.(1): the sum over the spin degree of freedom adds a factor of 2 for the U-term (the standard Coulomb interaction). Just wanted to verify this is a deliberate choice.

2- the authors write: "One can see that the biquadratic exchange can reach small positive values for $E\sim 2$ for frequencies $\omega=19$ and $25$. However, a similar feature is not observed for $\omega=35$.": upon zooming in, it seems to me that the solid blue line in Fig. 3 also becomes positive around $\mathcal{E}=3$?

3- caption to Fig 4: "White straight lines represent the frequencies for which $B_{ex}(\mathcal{E},\omega)$ diverges.": does the divergence happen for isolated frequencies with the current apparent finite width due to the frequency discretization used, or for entire frequency ranges? How would this resonance shift if one goes beyond the two-site approximation?

4- the authors write: "Therefore, to observe the hybridization, the electric field envelope must be changed slowly (adiabatic change)": how detrimental is this for heating effects in the full many-body system, where the spin and charge sectors are likely to broaden in energy and the off-resonant drive may turn resonant? In general, when a many-body system spends long time on resonance (i.e. traverses the avoided crossing slowly) the system absorbs energy and heats up.

5- the authors write: "It has been shown that the effective Hamiltonian picture can break down for long-time dynamics in the thermodynamic limit, because the system heats up to infinite temperature [40]. However, here we restrict ourselves to a two-site system to mimic the dynamics of a large system at relatively short timescales, for which heating does not occur.": heating is no longer exponentially delayed and can even occur at short times, when a parameter is changed slowly in the presence of the periodic drive, because the Adiabatic limit in Floquet systems does bot exist. The intuitive reason for this is that heating in driven systems occurs through rare Floquet resonances at high frequencies, which is why it takes exponentially long [more precisely the hybridization strength is exponentially suppressed with $\omega$]. However, adiabatic tuning of parameters on top of the drive typically forces the system to cross through such Floquet resonances thereby exacerbating the effect (see, e.g. arXiv:1606.02229).

6- the authors write: "In contrast to a common charge excitation by resonant photo-absorption, the spin-charge coupling allows non-resonant and reversible coupling to charge degrees of freedom.": while this is true for the two-site system, is reversibility expected to survive in the many-body case, i.e. beyond the two-site approximation? The model is certainly ergodic which, naively thinking, would cause the creation of entropy.

7- Is there a cold-atom realization of the two-site two-orbital model? Typically cold atoms can isolate double wells with pretty good efficiency. Additionally, one may be able to invoke additional hyperfine states to account for the orbital degrees of freedom? This may be a straightforward way to probe the reported effect in experiments.

Typos/Suggestions:

i- Sec 3.1, p8: "second canonical transformation in order project out states" --> "second canonical transformation in order TO project out states"

ii- in the dot-products of the spin-spin interaction. The authors may wish to use the latex command \cdot instead.

iii- denoting both time and hopping by $t$ is a bit inconvenient. It is the authors' choice how to label their quantities, yet if they see an alternative it might be worth considering.

Requested changes

1- In the definition of the vector potential $A_{ij}(t)$, the authors use $E_0$ to denote the electric field strength, but in some cases later on in the paper $\mathcal{E}$ is used instead. A definition appears right after Eq.(38). For completeness, it might be worth showing it also in the beginning, where the vector potential is first defined.

2- the authors write: "The exchange interactions have been computed for frequencies $\omega=19, 25\ and\ 35$...": is this in units of the bare hopping $t(0)$? Please put in the relevant energy scale throughout the paper.

  • validity: high
  • significance: high
  • originality: high
  • clarity: high
  • formatting: excellent
  • grammar: excellent

Author Marion Barbeau on 2019-02-04
(in reply to Report 1 by Marin Bukov on 2018-04-28)
Category:
answer to question

1- We added the definition of $\mathcal{E}$ earlier in the manuscript (Section2.1). 2- In the same section, we indicated that $t_0=1$ throughout the paper.

Author Marion Barbeau on 2019-02-01
(in reply to Report 1 by Marin Bukov on 2018-04-28)
Category:
answer to question

Response to referee 1

The manuscript "Optical control of competing exchange interactions and coherent spin-charge coupling in two-orbital Mott insulators" by Barbeau et al, studies the periodically-driven two-orbital Fermi Hubbard model. The paper derives the leasing-order effective Hamiltonian for the problem in the limit of strong interactions. The authors find that the presence of orbital dynamics and degrees of freedom does not change the control of the effective exchange interaction to leading order in the inverse interaction strength. At the same time, orbital dynamics gives access to charge states that are not accessible in equilibrium. This is demonstrated numerically in a two-site system using a Gaussian pulse signal. I recommend publication in SciPost, once the following suggestions/questions have been considered.

We thank the referee for the positive assessment of our manuscript as well as the comments and interesting questions raised in the report. The question 2 helped us to discover a systematic mistake in sign of the off diagonal matrix elements of the Hamiltonian matrix. We have recalculated the figures which led to quantitative difference. Importantly, all qualitative conclusions remain the same. In the following, we answer point-by-point the questions and comments from the referee.

1- In Eq.(1): the sum over the spin degree of freedom adds a factor of 2 for the U-term (the standard Coulomb interaction). Just wanted to verify this is a deliberate choice.

Here the summation is done only over the orbital degree of freedom ($\alpha$, $\beta$), but not over the spin degree of freedom ($\sigma$). Hence, there is no additional factor 2 from summing the spin degree of freedom.

2- the authors write: « One can see that the biquadratic exchange can reach small positive values for E~ 2 for frequencies w=19 and 25. However, a similar feature is not observed for w=35. »: upon zooming in, it seems to me that the solid blue line in Fig. 3 also becomes positive around E=3?

Due to the question 2 of referee 1, we re-analyzed the biquadratic exchange formula and discovered a mistake both in the analytical formula and in the exact Floquet spectra that we used to verify our results (the latter are not shown in the manuscript). Both errors consistently relate to the implementation mistakes in the minus signs of the off-diagonal elements in the Fourier representation of the hopping terms. The corrected results are shown in Figure 3a and the change of sign of the biquadratic exchange is indeed observed for all frequencies. To provide a more detailed explanation, we listed all distinct contributions to $B_{\textrm{ex}}$ (Eq (41-46)) and show their individual contribution as a function of driving strength in the new Figure 3b.

3- caption to Fig 4: «  White straight lines represent the frequencies for which Bex diverges. » : does the divergence happen for isolated frequencies with the current apparent finite width due to the frequency discretization used, or for entire frequency ranges? How would this resonance shift if one goes beyond the two-site approximation?

Within the canonical transformation, these resonances are sharp ($C_{dd’}^{νν’,m}$ terms which are proportional to $1/(E - m\omega)$). The reason is that we derive the canonical transform based on the strong coupling expansion, where E is given by the electronic energies without hopping terms. The width of the straight lines does not depend on the frequency discretization used to calculate this plot. Instead, it indicates approximately the regions where we expect the canonical transformation to be less accurate. From simulations in the single-orbital model, it is known that the canonical transformation breaks down not only at the resonances but is also less accurate in the vicinity of it [J.H. Mentink et al., Nat. Commun. 6, 6708 (2015)].

The relevant energy range in which strong deviation from the canonical transform is observed can be estimated from the bandwidth. For the highest excited state, the bandwidth is very small since four hoppings are required to couple different states in the $P_2^0$ sector. For states in the $P_1$ sector broadening occurs by a single hopping leading to a larger region where calculations beyond the strong-coupling expansion are needed.

4- the authors write: "Therefore, to observe the hybridization, the electric field envelope must be changed slowly (adiabatic change)": how detrimental is this for heating effects in the full many-body system, where the spin and charge sectors are likely to broaden in energy and the off-resonant drive may turn resonant? In general, when a many-body system spends long time on resonance (i.e. traverses the avoided crossing slowly) the system absorbs energy and heats up.

First of all, we emphasize that the driving frequency is not resonant. Therefore, no single-photon absorption occurs in the coupling to the charge sector $P_2^0$. Moreover, we expect that in extended systems, the charge band will not broaden so much since four hoppings are required to couple states within the $P_2^0$ sector. Therefore, we do not expect much heating to take place at the spin-charge coupling frequency with adiabatic tuning of the electric field.

5- the authors write: "It has been shown that the effective Hamiltonian picture can break down for long-time dynamics in the thermodynamic limit, because the system heats up to infinite temperature [40]. However, here we restrict ourselves to a two-site system to mimic the dynamics of a large system at relatively short timescales, for which heating does not occur.": heating is no longer exponentially delayed and can even occur at short times, when a parameter is changed slowly in the presence of the periodic drive, because the Adiabatic limit in Floquet systems does not exist. The intuitive reason for this is that heating in driven systems occurs through rare Floquet resonances at high frequencies, which is why it takes exponentially long [more precisely the hybridization strength is exponentially suppressed with w]. However, adiabatic tuning of parameters on top of the drive typically forces the system to cross through such Floquet resonances thereby exacerbating the effect (see, e.g. arXiv:1606.02229).

We agree with the referee that for a generic system, the adiabatic limit of Floquet may not exists. However, here the use of Floquet restricts to the derivation of an effective Hamiltonian which gives a qualitative picture of the avoided-crossing. We then confirm the reversibility of the spin-charge coupling effect using time-evolution of the full two-orbital Hubbard Hamiltonian (which does not rely on Floquet) under adiabatic tuning of the driving strength. Therefore, we conclude that no heating occurs in our system despite the fact that adiabatic tuning of parameters across the Floquet resonances may lead to increased heating [P. Weinberg et al., Phys. Rep. X 688 (2017)].

6- the authors write: "In contrast to a common charge excitation by resonant photo-absorption, the spin-charge coupling allows non-resonant and reversible coupling to charge degrees of freedom.": while this is true for the two-site system, is reversibility expected to survive in the many-body case, i.e. beyond the two-site approximation? The model is certainly ergodic which, naively thinking, would cause the creation of entropy.

We expect the charge sector we study will remain isolated from the rest of the spectrum even beyond the two-site system, since the broadening is small due to the large number of hoppings required to couple the different states. Therefore, the reversibility of the spin-charge coupling phenomenon might be possible also in extended systems.

7- Is there a cold-atom realization of the two-site two-orbital model? Typically cold atoms can isolate double wells with pretty good efficiency. Additionally, one may be able to invoke additional hyperfine states to account for the orbital degrees of freedom? This may be a straightforward way to probe the reported effect in experiments.

As we indicate in the outlook, we indeed anticipate that the spin-charge coupling effect can be studied in cold atom systems, where two-orbital two-site systems nowadays can be realized (see the cited references 45 and 46).

---

## Round 1 · Referee Report · Anonymous · 2018-4-29

Strengths

1. Careful development of the method
2. New and interesting emergent spin-orbit coupling from extended local Hilbert space

Weaknesses

1. An added discussion on the nature of inter-orbital coupling in terms of symmetries of the atomic orbitals would have added greatly to the work.

Report

The authors have studied a 2-orbital Hubbard model in a periodic electric field. The model is studied at half-filling in the strong coupling regime on a two-site system. Using a Floquet-type formalism, they derive the effective spin-exchange interactions up to 4th order in the hopping matrix element. Their results show that in addition to the usual bilinear-biquadratic interactions that are analogous to the single band results, the extended local Hilbert space results in a unique spin-orbit coupling term that is not possible in the single band counterpart. Finally, they compare their analytical results with direct numerical simulation of the model on a 2-site system.

The problem is interesting and of great current interest. AS the authors correctly point out, several real systems exist that are correctly modeled by a two-orbital model. The methods are well developed and there is no reason to doubt the validity of the results. The results obtained are interesting and add new phenomena achieved due to the extended local Hilbert space. The manuscript is fairly well written and deserves to be published. However, I would like the authors to consider the following.

My main critique of the work is that there is no discussion on the nature of the two orbitals. In many cases where two-orbital model is relevant, the orbitals come from different atomic orbitals that carry different angular momenta. This imposes selection rules for any transition between them. In the present work, it is implicitly assumed that both the orbitals are s-orbitals. Can the authors justify this assumption?

It would be nice to extend the results to a linear chain. Can the authors study a spin chain using the effective interactions derived? This can be compared with direct numerical simulation – with currently available computational power, I believe diagonalising an 8-site chain (with extensive use of symmetries) should be possible. Can this produce a Haldane phase? Can the relative strength of biquadratic term be varied enough to drive a transition away from the Haldane phase? Can the additional spin-orbit coupling induce new topological order? However, this is more of a suggestion.

Some other minor points:

1. Eq. (24) is not clear. Can the authors explain the symbols?
2. Use of m to denote both spin sector and Fourier components is a bit confusing. It will be better if the authors can use different symbols.
3. It would be good to provide an approximate dependence of the strength of the biquadratic term on the strength (amplitude) of the electric field

Requested changes

Some other minor points:

1. Eq. (24) is not clear. Can the authors explain the symbols?
2. Use of m to denote both spin sector and Fourier components is a bit confusing. It will be better if the authors can use different symbols.
3. It would be good to provide an approximate dependence of the strength of the biquadratic term on the strength (amplitude) of the electric field

  • validity: high
  • significance: high
  • originality: high
  • clarity: high
  • formatting: excellent
  • grammar: excellent

Author Marion Barbeau on 2019-02-01
(in reply to Report 2 on 2018-04-29)

Response to referee 2

The authors have studied a 2-orbital Hubbard model in a periodic electric field. The model is studied at half-filling in the strong coupling regime on a two-site system. Using a Floquet-type formalism, they derive the effective spin-exchange interactions up to 4th order in the hopping matrix element. Their results show that in addition to the usual bilinear-biquadratic interactions that are analogous to the single band results, the extended local Hilbert space results in a unique spin-orbit coupling term that is not possible in the single band counterpart. Finally, they compare their analytical results with direct numerical simulation of the model on a 2-site system. The problem is interesting and of great current interest. As the authors correctly point out, several real systems exist that are correctly modeled by a two-orbital model. The methods are well developed and there is no reason to doubt the validity of the results. The results obtained are interesting and add new phenomena achieved due to the extended local Hilbert space. The manuscript is fairly well written and deserves to be published. However, I would like the authors to consider the following.

We thank the referee for the positive assessment of our manuscript. Our point-by-point reply to the points to be considered is given below.

  • My main critique of the work is that there is no discussion on the nature of the two orbitals. In many cases where two-orbital model is relevant, the orbitals come from different atomic orbitals that carry different angular momenta. This imposes selection rules for any transition between them. In the present work, it is implicitly assumed that both the orbitals are s-orbitals. Can the authors justify this assumption?

Indeed, we do not mention the nature of the two orbitals in the manuscript. Such two-orbital models are derived as $e.g.$ a subset of d-orbitals in the presence of crystal field which occurs in transition-metal oxides [see for example A. Georges et al., Annual Review Cond. Mat. Phy. 4,137-178 (2013)]. We have added a comment to this in the model section. Since both orbitals originate from $e_g$ orbitals, no on-site electric dipole transition are allowed.

  • It would be nice to extend the results to a linear chain. Can the authors study a spin chain using the effective interactions derived? This can be compared with direct numerical simulation – with currently available computational power, I believe diagonalising an 8-site chain (with extensive use of symmetries) should be possible. Can this produce a Haldane phase? Can the relative strength of biquadratic term be varied enough to drive a transition away from the Haldane phase? Can the additional spin-orbit coupling induce new topological order? However, this is more of a suggestion.

The extension of our work to larger systems such as a linear chain is very interesting and the 8-site chain suggested might be numerically feasible. However, the current manuscript focuses on the analytical derivation of the effective interactions in multi-orbital systems, rather than directly simulating the time evolution of extended systems in response to such perturbations. In addition, we feel that the thorough finite size analysis which is needed to extrapolate from the 8-site simulation to the infinite system, is beyond the scope of the manuscript.

With the current two site model, we found that the biquadratic exchange cannot exceed the equilibrium value of the Heisenberg exchange interaction (see Section 3.1). Nevertheless, we can see that for certain field parameters, we can achieve $B_{\textrm{ex}}/J_{\textrm{ex}}$ ~ 1 which would be in the regime of the Haldane phase and/or dimerized phase. To address this, we included a short discussion on reaching non-trivial topological phases in the ground state of the effective Hamiltonian under driving, provided that a dynamic path between the initial state and this state can be realized, see Section 3.1. However, we emphasize that our interest is rather on studying reversible dynamics, which we also emphasized more explicitly in the introduction. It is an interesting yet open question, to what extend unitary dynamics under the effective Hamiltonian can bring a system from the antiferromagnetic phase close to the ground state of the new Hamiltonian. Such transition can be blocked due to constrains set by conserved quantities (see the example for changing the sign of the exchange interaction in the single-orbital model: Nat. Commun. 6(6708) (2015)). Similar constraints can prevent transitions to topologically non-trivial phases.

In the present work we restricted to a spin system therefore, we did not study the influence of spin-orbit coupling on the system. This is justified for $e_g$ orbitals, for which spin-orbit coupling matrix elements between different orbitals vanish.

Author Marion Barbeau on 2019-02-04
(in reply to Report 2 on 2018-04-29)

1- We added further explanations of the symbols used. 2- We changed the notation for the spin sector to $M_S$. 3- We added a new figure 3b which explicitly shows the dependence of different components of the biquadratic exchange on the strength of the electric field.

---

## Round 1 · Referee Report · Anonymous · 2018-5-8

Strengths

The manuscript provides complete details of the formalism used and is clearly written.

Weaknesses

The manuscript is very technical specially sections 2 and 3.

Report

In this manuscript, the authors study two-orbital Hubbard model at half filling under the action of a time-periodic electric field. The manuscript is an interesting extension of earlier work ( Nature Communications {\bf 6}, 6708 (2015)), coauthored by two of the authors, where effect of a time periodic electric field on single orbital Hubbard model was studied.
Further the Schriefer-Wolf transformation for mulifavour Hubbard model has been developed in detail for the static case where there is no time dependent term in the Hamiltonian (Ref (21) in the manuscript). Hence there is no doubt on the approach used in this manuscript.

But it also means the main new contribution of the current manuscript is to combine the works of Ref(21) and the earlier mentioned reference on time periodic electric field and understand the physics of two-orbital Hubbard model in the presence of time periodic electric field. This brings in some new features like the tuning of biquadratic exchange interactions due to the time periodic electric field and the extension of physics beyond the effective Hamiltonian of spin-one model. I am ready to support the publication of this contribution, if the authors address the following issues.
1. Authors should provide discussion on the possible phases and phase diagram that can emerge in the presence of time periodic electric field in two-orbital Hubbard model and provide a comparison with the phase diagram of single orbital case in the presence of time periodic electric field.

2. It will be good to provide a physical mechanism which results in change in sign of the spin exchange coupling and the biquadratic coupling in the presence of time periodic electric field as shown in Fig 3 of the manuscript.
3. Authors do exact numerical simulation for 2 site Mott Hubbard model. It will be good to solve the model for a 1d chain numerically, going upto 8 or 10 sites should be possible with current day computational facilities.

4. Authors should try to solve the effective low energy Hamiltonian obtained, at least a mean field solution of effective low energy Hamiltonian should be provided to make the physics coming out of this approach more explicit.

5. Though the manuscript has been written very clearly, it is very technical and hard for readers to grasp. Therefore, I urge the authors to offer a more detailed discussion of the physics emerging from this work, as to allow the reader to appreciate what is learned from this work. Comparison of results (at least within mean field) with those for static two orbital Hubbard model and dynamical(that is, in the presence of time periodic electric field) single orbital Hubbard model would also be useful.
6. It will be good to discuss at least briefly what changes are expected in the emerging physics if the electric field is not a periodic function of time but some aperiodic or quasiperiodic function of time.

Requested changes

Changes required based on physics discussed in the manuscript are mentioned in the report.

  • validity: good
  • significance: ok
  • originality: ok
  • clarity: good
  • formatting: good
  • grammar: good

Author Marion Barbeau on 2019-02-01
(in reply to Report 3 on 2018-05-08)

Response to referee 3

In this manuscript, the authors study two-orbital Hubbard model at half filling under the action of a time-periodic electric field. The manuscript is an interesting extension of earlier work ( Nature Communications 6, 6708 (2015)), coauthored by two of the authors, where effect of a time periodic electric field on single orbital Hubbard model was studied. Further the Schriefer-Wolf transformation for mulifavour Hubbard model has been developed in detail for the static case where there is no time dependent term in the Hamiltonian (Ref (21) in the manuscript). Hence there is no doubt on the approach used in this manuscript. But it also means the main new contribution of the current manuscript is to combine the works of Ref(21) and the earlier mentioned reference on time periodic electric field and understand the physics of two-orbital Hubbard model in the presence of time periodic electric field. This brings in some new features like the tuning of biquadratic exchange interactions due to the time periodic electric field and the extension of physics beyond the effective Hamiltonian of spin-one model. I am ready to support the publication of this contribution, if the authors address the following issues.

We thank the referee for positive assessment of our manuscript and give our reply to the issues mentioned below.

1- Authors should provide discussion on the possible phases and phase diagram that can emerge in the presence of time periodic electric field in two-orbital Hubbard model and provide a comparison with the phase diagram of single orbital case in the presence of time periodic electric field.

To address more explicitly the possible phases that can emerge from dynamically perturbed exchange interactions we have included examples in the introduction, both for quantum spin chains and classical spins. However, we also emphasize that the insights obtained from the study of equilibrium phase diagrams give limited insight on the short time-dynamics which emerges after a non-resonant driving. For instance, the change of sign of $J_{\textrm{ex}}$ gives rise to an effective time reversal rather than a transition from antiferromagnetic to ferromagnetic phase [J.H. Mentink et al., Nat. Commun. 6(6708) (2015)].

Instead, our aim is to study the reversible control of exchange under unitary time-evolution, as we mentioned more explicitly in the introduction. Nevertheless, the study of the effect of competing exchange interactions on the magnetic order is an interesting and computationally challenging prospect that is left for future work.

2- It will be good to provide a physical mechanism which results in change in sign of the spin exchange coupling and the biquadratic coupling in the presence of time periodic electric field as shown in Fig 3 of the manuscript.

The physical mechanism for the change of sign of Jex and Bex is similar as what has been discussed in Nat. Commun., 6, 6708 (2015) for single-orbital model. This is, photo assisted hoppings lead to competing effects (i) coherent destruction of tunneling that reduces the normal exchange path and (ii) virtual excitations to states with an energy shifted by $U + m\omega < 0$ give contributions of opposite sign. Together, this can lead to a change of sign of $J_{\textrm{ex}}$ and $B_{\textrm{ex}}$ for certain frequency and driving strength >1 where the normal exchange path is strongly reduced. For Bex there are several distinct exchange paths which already in equilibrium partially compensate each other. To clarify thus, we have added a more elaborate description of these various path in Section 3.1 and display their quantitative contribution in a new Figure 3b.

3- Authors do exact numerical simulation for 2 site Mott Hubbard model. It will be good to solve the model for a 1d chain numerically, going up to 8 or 10 sites should be possible with current day computational facilities.

Indeed, the extension of our current work to larger systems such as a linear chain is very interesting and the 8-site chain suggested might be numerically feasible. However, the current manuscript focuses on the analytical derivation of the effective interactions in multi-orbital systems, rather than directly simulating the time evolution of extended systems in response to such perturbations. In addition, 8-sites system remains finite therefore, it is unclear what can be learned about phase transitions in infinite 1D systems.

4- Authors should try to solve the effective low energy Hamiltonian obtained, at least a mean field solution of effective low energy Hamiltonian should be provided to make the physics coming out of this approach more explicit.

To address this point we make explicit reference to existing classical calculations of systems with additional biquadratic exchange, which lead for example to spin spiral phases. To make the consequences of the physics more clear, we also extended the discussion about the physics behind the control of Bex (see Section 3.1). In addition, we added a discussion on possible quantum phases in 1D spin chains which could arise in the regime where $B_{\textrm{ex}}/J_{\textrm{ex}}$ ~ 1 [see Section 3.1]. However, because the short-time evolution under the effective Hamiltonian (which is addressed in the first place within the Floquet approach) is different from the equilibrium phase, we feel that the calculation of a mean-field phase diagram is beyond the scope of this work.

5- Though the manuscript has been written very clearly, it is very technical and hard for readers to grasp. Therefore, I urge the authors to offer a more detailed discussion of the physics emerging from this work, as to allow the reader to appreciate what is learned from this work. Comparison of results (at least within mean field) with those for static two orbital Hubbard model and dynamical (that is, in the presence of time periodic electric field) single orbital Hubbard model would also be useful.

We thank the referee for the suggestions. The present work focuses on the control of exchange interactions in two-orbital systems out of equilibrium. We already explicitly compare non-equilibrium Heisenberg exchange interaction in the single and two-orbital model (Eq. (38,39)) which are very similar.

The additional biquadratic exchange gives rise to effects not present in the single-band model. For example, in quantum chains it can lead to stabilization of the Haldane phase if the biquadratic exchange is strong enough. In addition, for classical spins a non-collinear spin structure is favored for positive $B_{\textrm{ex}}$, which competes with the Heisenberg exchange. The mean-field analysis of these effects have been discussed before [D.A. Yablonskii, Physica C: Superconductivity, 171, 5–6 (1990)] and recently also the nonequilibrium response to changes of the ratio of $J_{\textrm{ex}}$ and $B_{\textrm{ex}}$ have been studied on the basis of classical atomistic spin dynamics simulations [J. Hellsvik J.H. Mentink and J. Lorenzana, Phy. Rev. B 94, 144435 (2016)]. Therefore, we did not analyze this further in the present manuscript. However, to give the reader a better understanding of the physics that emerges from the presence of both $J_{\textrm{ex}}$ and $B_{\textrm{ex}}$, we added a short discussion in Section 3.1.

We think that the main novelty learnt from our work is the emergence of coupling between spin and charge degrees of freedom (see section 3.2 « Beyond the spin-one model »). This is specific to multi-orbital systems and appears only out of equilibrium. In the Conclusion, we further elaborate that, besides the possibility to induce coherent charge dynamics, the presence of the spin-charge coupling should also be visible for short pulses, enabling the excitation of doubly ionized states which could remain coherent due to the gapping with the normal Mott-Hubbard gap.

6- It will be good to discuss at least briefly what changes are expected in the emerging physics if the electric field is not a periodic function of time but some aperiodic or quasiperiodic function of time.

We agree that it is very interesting to analyze what control is feasible beyond periodic driving. We emphasize that In Section 4 we already study a Gaussian envelop in addition to a periodic field. Our approach with the canonical transformation can also be applied for arbitrary time-dependent fields, as we indicated in Sec. 2.3 of the manuscript citing recent work of some of us [M. Eckstein et al., arXiv:1703.03269v1 (2017)] . Also, control with DC fields has been done already, also for two-orbital models, see [K. Takasan and M. Sato, arXiv:1802.04311v1 (2018)], which we now include in the references. It would indeed be interesting to further extend our approach to more exotic forms of time-dependent fields, which we leave for future work.

---

## Round 2 · Referee Report · Marin Bukov (Referee 1) · 2019-2-16

Report

The paper quality improved, small gaps in the exposition were filled, and figures -- improved. The authors also reflected in satisfactory manner all my questions in the new version of the manuscript.

I recommend publication without further delay.

---

## Round 2 · Author Response

We thank the referees for their comments and suggestions which helped to further improve the manuscript. We provide a detailed explanation on list of changes in the designated section below.

---

## Round 2 · List of Changes

Following the remarks from the referees, we expanded the analysis of the biquadratic exchange interaction and we added of a new figure 3b. Moreover, Figure 2 has been modified in order to better illustrate the different canonical transformations used to derive the exchange interactions. In addition, to make the physics emerging from our work more clear, we added a short discussion on the possible phases that could emerge from the ground state of the effective spin model under driving. In passing through this analysis we unfortunately found a systematic error in signs of the off-diagonal elements of the Hamiltonian matrix, which we have corrected. Therefore, all figures have been recalculated. Importantly, while this leads to small quantitative differences, all qualitative conclusions remain the same.

---

## Editorial Decision

published